# Variability of the geothermal gradient across two differently aged magma-rich continental rifted margins of the Atlantic Ocean: The Southwest African and the Norwegian margins

Ershad Gholamrezaie[1,2], Magdalena Scheck-Wenderoth[2,3], Judith Sippel[2], and Manfred R. Strecker[1]

[1]Institute of Earth and Environmental Science, University of Potsdam, Germany
[2]Helmholtz Centre Potsdam–GFZ German Research Centre for Geosciences, Postdam, Germany
[3]Faculty of Georesources and Material Engineering, RWTH Aachen, Aachen, Germany

*Correspondence to:* Ershad Gholamrezaie (ershad@gfz-potsdam.de)

**Abstract.** The aim of this study is to investigate the shallow thermal field differences for two differently aged passive continental margins by analyzing regional variations in geothermal gradient and exploring the controlling factors for these variations. Hence, we analyzed two previously published 3D conductive and lithospheric-scale thermal models of the Southwest African and the Norwegian passive margins. These 3D models differentiate various sedimentary, crustal and mantle units and integrate different geophysical data such as seismic observations and the gravity field. We extracted the temperature-depth distributions in 1 km intervals down to 6 km below the upper thermal boundary condition. The geothermal gradient was then calculated for these intervals between the upper thermal boundary condition and the respective depth levels (1, 2, 3, 4, 5, and 6 km below the upper thermal boundary condition). According to our results, the geothermal gradient decreases with increasing depth and shows varying lateral trends and values for these two different margins. We compare the 3D geological structural models and the geothermal gradient variations for both thermal models and show how radiogenic heat production, sediment insulating effect, and thermal Lithosphere-Asthenosphere Boundary (LAB) depth influence the shallow thermal field pattern. The results indicate an ongoing process of oceanic mantle cooling at the young Norwegian margin compared with the old SW African passive margin that seems to be thermally equilibrated at the present-day.

## 1 Introduction

Comprehension of the lithosphere-scale thermal state is a key to unraveling the evolution, strength, and physical and chemical processes of the lithosphere (e.g., Davies, 1980; Chapman, 1986; Artemieva and Mooney, 2001; Scheck-Wenderoth and Lamarche, 2005; McKenzie et al., 2005; Ebbing et al., 2009). Furthermore, analyzing the thermal field of the lithosphere has important applications in industrial sectors such as geo-resources exploration (e.g., Muffler and Cataldi, 1978; Tissot et al., 1987; Grevemeyer and Villinger, 2001; Wallmann et al., 2012).

The lithospheric thermal field generally depends on the thermal thickness and the thermal properties of the lithosphere. This has been deduced from continental crustal geotherm (Pollack, 1986; McKenzie and Bickle, 1988; Rudnick and Nyblade, 1999; Kaminski and Jaupart, 2000; Artemieva and Mooney, 2001; Artemieva, 2006; Jaupart and Mareschal, 2007; Mareschal

and Jaupart, 2013) and from plate cooling models explaining oceanic heat flow pattern and seafloor depth evolution (Parsons and Sclater, 1977; Johnson and Carlson, 1992; Stein and Stein, 1992; Goodwillie and Watts, 1993; DeLaughter et al., 1999; Watts and Zhong, 2000; Crosby et al., 2006; Crosby and McKenzie, 2009). There is a consensus that conduction is the main heat transfer mechanism in the lithosphere and generally controlled by (1) the heat input from larger mantle depths, (2) the heat internally produced in the lithosphere by the decay of radioactive elements, and (3) the thermal conductivity of different lithospheric layers (Summary in Allen and Allen, 2005; Turcotte and Schubert, 2014). The interaction of these controlling factors complicates predictions of temperature increase with depth. This difficulty largely arises from the variability of the lithosphere in terms of structure and composition, parameters that are a product of the tectonic setting and evolution of the location of interest. One well-established strategy to investigate the present-day thermal field of a certain area is to integrate existing geophysical and geological data into 3D structural models that provide the basis for numerical modeling, which simulates heat transport processes after setting boundary conditions and thermal properties according to the geological structure (e.g., Scheck-Wenderoth and Lamarche, 2005; Noack et al., 2013; Scheck-Wenderoth et al., 2014; Sippel et al., 2015; Balling et al., 2016).

Although there is already a large number of such 3D models for different settings worldwide, none of these studies has focused on the variability of geothermal gradients with respect to geological structure. It is self-evident that the geothermal gradient is a function of local temperature, which depends on the thermal state. This is an important issue, because, geothermal gradient variations in the shallow parts of the subsurface (measured or modeled) may be indicative of the thermal field and temperature-controlled processes at greater depths. Moreover, methods used to assess the thermal history of specific tectonic settings generally apply strongly simplified assumptions concerning the geothermal gradient and its changes in space and time (e.g., Burnham and Sweeney, 1989; Barker, 1996; Allen and Allen, 2005; Naeser and McCulloh, 2012). Accordingly, a better understanding of variations of the geothermal gradient could also improve the quality of thermochronological results of thermal history models. In this interest, the main questions are: (1) how does the geothermal gradient vary with depth and laterally over major geological structure (such as passive continental margins)?; (2) what are the controlling factors of these variations?; and (3) how are shallow geothermal gradients related to the lithospheric-scale thermal field?

The goal of this study is to investigate geothermal gradient as one manifestation of the thermal field that can directly be observed and usually differs significantly according to the specific tectonic settings. Concerning thermal histories, we do not go into much detail as we do not reconstruct thermal histories. The point we want to make is to raise awareness in the context of paleo-thermal conditions. Our approach follows three principal steps: (1) derive geothermal gradients from two existing and validated 3D thermal models, both from volcanic passive margins, but with major age differences: the SW African passive margin (130 Ma) and the Norwegian margin (55 Ma); (2) investigate the variability of geothermal gradients with respect to the structural configuration changing from unthinned continental lithosphere onshore, over the stretched margins with great sediment thickness, and finally to the distal oceanic lithosphere; and (3) compare the results of the calculated geothermal gradients for the two different margins. In this context, there are significant variations in the thermal field that need to be considered, when sediments, crust, and the lithospheric mantle display pronounced lateral heterogeneities in thickness and composition across the continental margins. In spite of a very similar configuration of the crust, the underlying lithospheric

mantle in the two study areas differs. The younger lithospheric mantle beneath the oceanic crustal parts of the North Atlantic is significantly thinner than the older counterpart of the South Atlantic (Scheck-Wenderoth et al., 2007; Scheck-Wenderoth and Maystrenko, 2008; Maystrenko et al., 2013). By comparing the calculated geothermal gradients of these margins, we particularly address the consequences of the lateral heterogeneities for the thermal field, and test the hypothesis that the present-
5  day thermal field is different for the two settings and ultimately determined by the lithospheric mantle characteristics.

## 2  Method

### 2.1  3D Conductive Thermal Model

Theoretically, heat is transfered due to a temperature gradient and dependent on the thermal conductivity within the solid media. This statement is known as the law of heat conduction or Fourier's law [eq. 1], where $\lambda$ stands for the thermal conductivity,
and $\nabla T$ defines the premier temperature gradient;

$$\boldsymbol{q} = -\lambda \nabla T. \tag{1}$$

Considering Fourier's law (Eq.1) and assuming conductive heat transport as the main heat transfer mechanism, the heat flow equation can be derived on a lithospheric scale (Eq. 2). In these equations T and t represent temperature and time, respectively. The radiogenic heat production is shown by S and $\Delta$ is the Laplacian operator. The parameter $\rho$ stands for density, c for the
heat capacity and $\lambda$ for the thermal conductivity.

$$\rho c \frac{\partial T}{\partial t} = -\lambda \Delta T + S \tag{2}$$

The two considered 3D conductive thermal models (Scheck-Wenderoth and Maystrenko, 2008; Maystrenko et al., 2013) were created as a numerical solution to Eq. 2 in the steady-state condition ($\frac{\partial T}{\partial t} = 0$) and by considering lithology-dependent thermal properties (Table 1). The lower thermal boundary in these models has been fixed at the 1300 °C isotherm signifying
the thermal LAB depth, whereas the topography-bathymetry surface with a constant temperature (Norway: 2 °C; SW Africa: 5 °C) has been set as the upper thermal boundary.

### 2.2  Geothermal Gradient

The geothermal gradient is the temperature change with increasing depth [Eq. 3]. Through the 3D thermal models, the corresponding temperature to a certain depth is predicted which simplifies the geothermal gradient calculation. However, since
the aim of this study is to compare the variations of the geothermal gradient in different geological settings, a comparable reference frame is required. Therefore, the upper thermal boundary in each thermal model was chosen as the reference surface. We extracted the temperature-depth distributions in 1 km homogeneous depth intervals down to 6 km below the upper thermal boundary surface (Fig. 1, see also Fig. 1 and 2 in the Supplement). To calculate the geothermal gradient (Eq. 3), we

**Table 1.** Oceanic lithosphere age after Müller et al. (2008) and average physical properties of geological units used for thermal modeling after Scheck-Wenderoth and Maystrenko (2008); Maystrenko et al. (2013).

| Geological units | SW African margin | | | | Norwegian margin | | | |
|---|---|---|---|---|---|---|---|---|
| | Age | Maximum thickness [km] | Average thermal conductivity [W/mK] | Average heat production [µW/m³] | Age | Maximum thickness [km] | Average thermal conductivity [W/mK] | Average heat production [µW/m³] |
| Clastic sediments | Cenozoic | Walvis Basin = 3.6 Lüderitz Basin = 3 Orange Basin = 1.4 | 1.5 | 1 | Cenozoic | Vøring Basin = 3.2 Møre Basin = 3.6 | 1.5 | 1 |
| | Cretaceous | Walvis Basin=5 Lüderitz Basin=8 Orange Basin=15 | 1.5 | 1 | Cretaceous | Vøring Basin = 16 Møre Basin = 13 | 1.5 | 1 |
| Upper crystalline crust | - | 50 | 2.8 | 1.45 | - | 40 | 2.7 | 0.8 |
| High density crust | - | - | 2.7 | 0.95 | - | - | 2.6 | 0.3 |
| High velocity body | - | - | 2.6 | 0.8 | - | - | 2.6 | 0.5 |
| Oceanic crust | 130 Ma b.p. | - | 2.75 | 0.3 | 55 Ma b.p. | - | 2.1 | 0.3 |
| Lithospheric mantle ($600°C \leq T \leq 1300°C$) | - | 135 | 3.95 | 0.03 | - | 110 | 3.95 | 0.03 |

considered "$T_j$" and "$z_j$" respectively as the temperature and the elevation of a surface in the 3D thermal models for which the upper thermal boundary condition was assigned to. In our calculation, "$z_i$" was the corresponding depth for $i = 1, 2, 3, 4, 5, 6$ km below the upper thermal boundary condition, and "$T_i$" was the temperature distributions at the corresponding depth levels of "$i$"s (Fig. 1, see also Fig. 1 and 2 in the Supplement). The geothermal gradient was then calculated for these intervals as the temperature difference between the uppermost surface and the corresponding depth levels. Thus, the average geothermal gradient is determined for increasingly thicker intervals with increasing depths.

$$\frac{dT}{dz} = \frac{T_i - T_j}{z_i - z_j} \tag{3}$$

We have chosen this way of illustrating the depth evolution of the geothermal gradient to make our assessment of average geothermal gradient variation comparable to the observation-derived geothermal gradient variation. In practice geothermal gradients are often calculated from surface heat flow and bottomhole temperature measurements. As therefore bottomhole temperatures depend on the absolute depth of the drilled well, the derived average geothermal gradients vary accordingly. Our goal was to show: (1) that there is no such a thing as one average geothermal gradient and (2) that the latter is subject to variation in response to depth and structural heterogeneity.

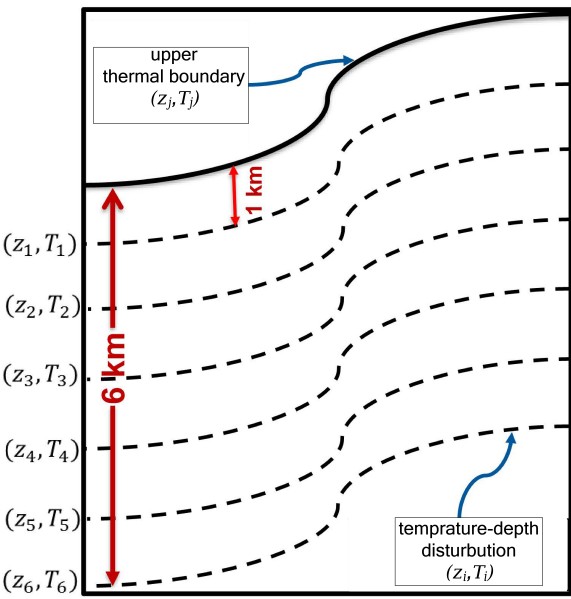

**Figure 1.** The geothermal gradient calculation: schematic of the temperature-depth distributions in 1 km homogeneous depth intervals down to 6 km below the upper thermal boundary surface to calculate the geothermal gradient between the uppermost surface ($z_j$,$T_j$) and the corresponding depth levels ($z_i$,$T_i$). The temperature-depth distribution maps are presented as figure 1 and 2 in the Supplement.

## 3 Exploited Models

The database for this study consists of regional 3D lithospheric-scale structural and thermal models for the SW African (Maystrenko et al., 2013) and the Norwegian passive margins (Scheck-Wenderoth et al., 2007; Scheck-Wenderoth and Maystrenko, 2008). These models integrate and are consistent with observed seismic data, gravity data, as well as with measured tempera-
tures and heat flow.

### 3.1 Geological Background

Passive continental margins evolve in consequence to continental rifting and breakup with the formation of new oceanic crust (White et al., 1987; Huismans and Beaumont, 2008). Rifted margins, according to the level of volcanism, are divided into two general categories: (a) magma-poor rifted margins and (b) magma-rich (volcanic) rifted margins (Franke, 2013). One of the typical characteristics of magma-rich passive continental margins is an only moderately thinned at the proximal margin (compared to magma-poor margins) whereas crustal thinning at the distal margin is significant. As their parts of the continental crust are replaced by lower crustal bodies, the remaining ordinary crystalline crust is thinned to a few km. These lower crustal bodies are usually characterized by high p-wave velocities of more than 7.3 km/s (White et al., 1987; Talwani and Abreu, 2000; Lavier and Manatschal, 2006; Huismans and Beaumont, 2008; Franke, 2013). Two more characteristic features of volcanic passive margins are Seaward Dipping Reflectors (SDRs; interpreted as the expression of basaltic extrusions (Hinz, 1981; Mutter et al., 1982; White et al., 1987; White and McKenzie, 1989)), and usually more than 10 km thick syn- and post-rift sediments (White and McKenzie, 1989; Huismans and Beaumont, 2008; Franke, 2013).

The volcanic passive margin of Norway is the result of the last phase of Pangea breakup (55 Ma b.p) in the early Cenozoic. It evolved in response to the North Atlantic breakup, presumably, initiated by the abnormally hot mantle of the Iceland plume (White, 1989; Skogseid et al., 1992; Ren et al., 1998). The Norwegian continental margin records several pre-breakup rifting phases that played a major role in initiating the formation of deep sedimentary basins (Skogseid, 1994; Blystad et al., 1995; Swiecicki et al., 1998; Doré et al., 1999, 2002). For instance, the deep Vøring and Møre basins in the Norwegian Sea were formed due to the early rifting around 150 Ma ago and contain more than 10 kilometers of late Mesozoic deposits (Scheck-Wenderoth et al., 2007).

In contrast, the causative breakup event leading to the creation of the South Atlantic and the formation of the SW African margin occurred significantly earlier than the Norwegian margin formation. About 130 Ma ago the continental lithosphere broke apart and generated the South Atlantic Ocean (Larson and Ladd, 1973; Rabinowitz and LaBrecque, 1979; Unternehr et al., 1988; O'Connor and Duncan, 1990; Nürnberg and Müller, 1991; Brown et al., 1995; Talwani and Abreu, 2000; Blaich et al., 2009). These processes were followed by rifting and post-breakup cooling, resulting in several sedimentary basins formed along the margins of the South Atlantic (Stewart et al., 2000; Macdonald et al., 2003; Séranne and Anka, 2005; Dressel et al., 2016).

Both passive margin settings have similar configurations; crystalline crustal rocks crop out onshore, thick sedimentary sequences along the rifted margins are underlain by a severely thinned upper crust and are associated with high-velocity high-

density lower crustal bodies, and display pronounced SDRs. The sedimentary units at both settings are predominantly composed of siliciclastic rocks with varying degrees of compaction (Stewart et al., 2000; Brekke, 2000; Scheck-Wenderoth et al., 2007; Scheck-Wenderoth and Maystrenko, 2008; Maystrenko et al., 2013). The crustal configuration of these sedimentary basins and their evolution during different tectonic phases are partly discussed controversly (Stewart et al., 2000; Macdonald et al., 2003; Fernandez et al., 2005; Lundin and Doré, 2011; Koopmann et al., 2014; Nirrengarten et al., 2014; Gernigon et al., 2015; Dressel et al., 2016; Mjelde et al., 2016; Maystrenko et al., 2017). However, all concepts agree with respect to the presence of seaward dipping reflections near the continent-ocean transition, of a thick sedimentary succession above thin crystalline crust beneath the margins and a high-velocity high-density lower crustal body below the distal margins. All studies furthermore agree that the North and South Atlantic oceans are of significantly different age. Controversies emerged with respect to geodynamic concepts explaining observed variations in subsidence rate and uplift phases during the post-rift evolution. In this context especially the nature of the lower crustal high-velocity high-density bodies and the role of mantle dynamics for post-breakup vertical movements are debated. For the margins along the South Atlantic the lower crustal bodies are predominantly interpreted as the relics of breakup-related mafic underplating (gabbros), whereas in the North Atlantic serpentinized mantle and eclogites as reminder of earlier orogenies, are discussed as alternative explanations (White and McKenzie, 1989; Eldholm et al., 2000; Gernigon et al., 2004; Ebbing et al., 2006). Autin et al. (2016) have examined the thermal implications of these different hypotheses for South Atlantic Argentine margin and concluded that only a serpentinite composition would imply a significantly colder thermal field, whereas eclogite and gabbro have similar thermal effects.

However, there are also some major differences between the two margins. Major differences are related to the different times of breakup and the different post-breakup histories. The younger N-Atlantic margin is bordered by a younger and thinner oceanic lithosphere and shows a thickened oceanic crust near the continent-ocean transition compared to the S-Atlantic margin.

For both margins, lithosphere-scale structural models and results from simulations of the steady-state conductive thermal field have been published (Scheck-Wenderoth and Maystrenko, 2008; Maystrenko et al., 2013). Though these thermal models have been produced using roughly the same workflow, there are specific differences with regard to their parametrization in response to the individual resolution and availability of data on thermal properties (Table 1). With this study we concentrate on spatial variations of the present-day thermal field response to first order differences in structural setting and related distribution of lithological units and their thermal rock properties.

### 3.2   Southwest African Passive Margin

There are three main sedimentary basins in the study area of the SW African continental margin. From North to South there are the Walvis Basin, the Lüderitz Basin, and the Orange Basin. These basins overlie a thinned continental crust and are filled with Cretaceous and Cenozoic sediments (Fig. 2a, Table 1). The Orange Basin hosts the thickest sediments compared to the two other basins with a maximum thickness of up to 16 km in the southern part of the basin. Sediment thickness varies in a similar manner in the Lüderitz and Walvis basins and ranges between 5 to 8 km except of small parts of the Walvis Basin, where up to 10 km of sediments are present. Onshore, the model also differentiates upper Proterozoic sediments (Owambo and Nama basins: (Clauer and Kröner, 1979; Miller, 1997)). The Continent-Ocean Boundary (COB; determined from gravity data

in combination with reflection seismic and magnetic data (Pawlowski, 2008)) runs approximately along the 5 km isopach of the sedimentary fill and parallel to the coastline.

Below the sedimentary basins, the top-crystalline basement descends seaward. Offshore, where the Walvis Ridge intercepts the coast, the shallowest basement is at a depth of 2000 meters below sea level (BSL). With 17 km BSL the top-basement is deepest in the South-Southeast beneath the Orange Basin (Fig. 2b). The upper crystalline crustal thickness is largest onshore, with a maximum thickness of more than 45 km. Towards the COB, the thickness of the crystalline crust progressively decreases and attains less than 5 km in the oceanic crustal domain (Fig. 3a).

The depth of the Moho varies between 20 and 30 km BSL beneath the continental shelf (Fig. 3b), where the lithospheric mantle (the layer between the Moho and the Lithosphere-Asthenosphere Boundary) has the largest and smallest thickness beneath the onshore area, with 135 and 75 km, respectively (Fig. 3c). Beneath the sedimentary basins, the thickness of the lithospheric mantle is approximately uniform and stays in the range between 80 and 100 km.

In their 3D thermal model, Maystrenko et al. (2013), considered a temperature of 5 °C as upper thermal boundary condition at surface and seafloor, respectively. The topography and bathymetry of these surfaces is displayed in figure 4a. The topography reaches a height of more than 1500 meters above sea level (ASL) and decreases seaward. Offshore, the continental shelf is a few hundred meters BSL; the continental slope descends steeply to the isobath of 3000 meters BSL at the COB. Farther, the seafloor descends with a more gentle slope to 5000 meters BSL. In the investigated area, the deepest part of the ocean is located in the southwestern corner of the model with a depth to 5500 meters BSL.

Along with the top surface and the seafloor, the Lithosphere-Asthenosphere Boundary (LAB) constitutes also a crucial element in the structural/thermal model as it is the interface to which the lower thermal boundary condition of 1300 °C is assigned. The LAB is deepest ($\sim$ -180 km) beneath the onshore areas in the Northeast and shallowest ($\sim$ -100 km) under the oceanic region (Fig. 4b). Beneath the sedimentary basins of the continental margin, the LAB is situated at a depth of 115 to 120 km BSL, except for the southern part of the Orange Basin, where the depth of the LAB descends to 130 km BSL.

## 3.3 Norwegian Passive Margin

The Norwegian passive margin includes the Vøring and the Møre basins. We extracted the cumulative thickness of sediment packages from the structural model (Fig. 5a). The thickest part of the sediments lies within the Vøring Basin, with a thickness of up to 17 km. Compared to the Vøring Basin, the sediments within the Møre Basin are thinner and rarely thicker than 12 km. The sedimentary thickness is more uniform along the COB and approximately follows the 8 km isopach.

Over the whole area, the depth to the crystalline basement varies between more than 1500 meters ASL to 18 km BSL (Fig. 5b). The deepest parts of the basement are located beneath the sedimentary basins, and parallel to the COB. The depth to the top of the crystalline basement is almost uniform below the oceanic crustal domain and varies between 5 to 6 km BSL. The thickness of the upper crystalline crust (Fig. 6a) is largest onshore, with more than 35 km beneath the Norwegian Caledonides. Offshore, the thickness decreases seaward to about less than 5 km in the oceanic crustal domain.

According to the crustal structure, the Moho is deeper (17 to 37 km BSL) below the continental crust compared to the oceanic crust where the Moho is inferred to be located at a depth of 9 to 20 km BSL (Fig. 6b). Below the Moho, the thickness

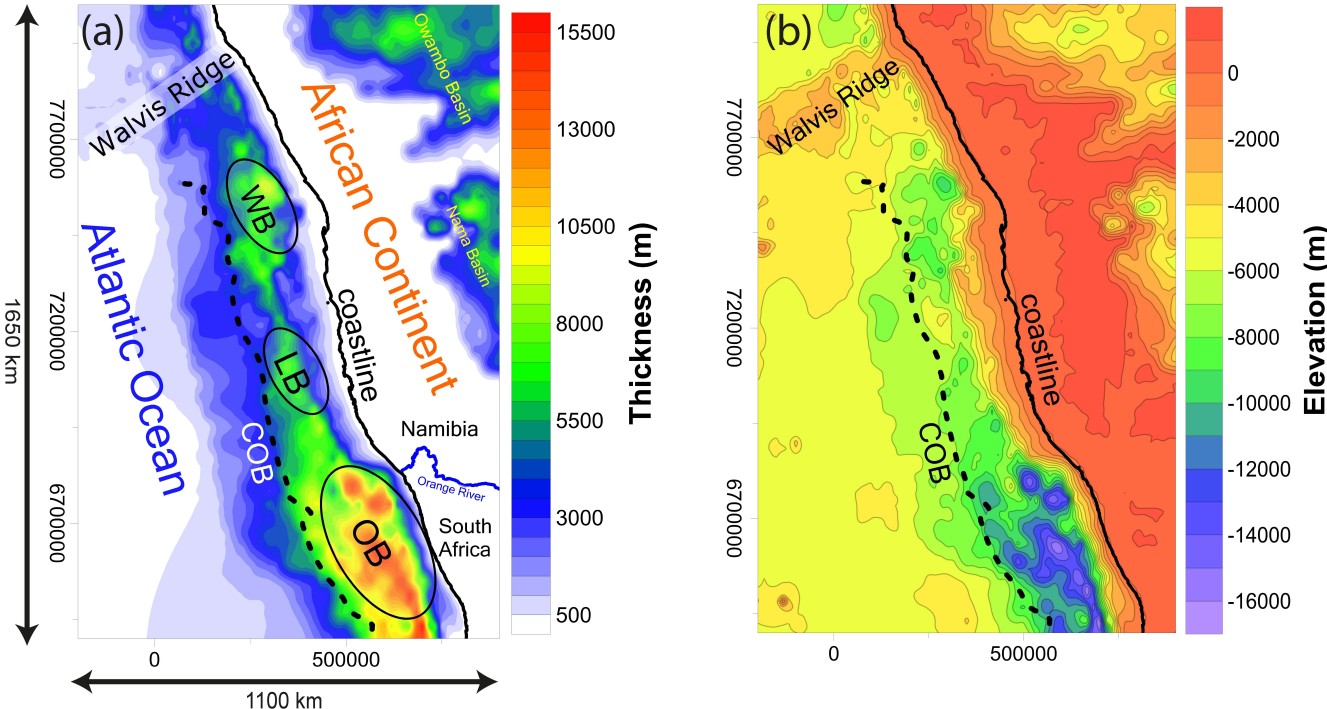

**Figure 2.** 3D structural model of the SW African passive margin: (a) cumulative thickness map of sediments including offshore Cretaceous-Cenozoic thick sedimentary basins and onshore Proterozoic Owambo and Nama basins; (b) depth to top-basement (COB: Continent-Ocean Boundary, WB: Walvis Basin, LB: Lüderitz Basin, OB: Orange Basin, UTM: WGS84, zone 33S).

of the lithospheric mantle decreases seaward from 110 km in the continental domain to 45 km in the oceanic crustal domain (Fig. 6c).

For the thermal model (Scheck-Wenderoth and Maystrenko, 2008), a 2 °C isotherm was assigned as the upper thermal boundary condition at the topography and bathymetry (Fig. 7a). Onshore, the topography reaches elevations of close to 2000 meters ASL and descends seaward. Offshore, the wide continental shelf is a few hundred meters BSL and descends to more than 3500 meters BSL in the oceanic crustal domain.

In addition to the upper thermal boundary condition setting and equivalent to the SW African margin, the LAB surface was considered as the lower thermal boundary condition (1300 °C). The depth to the LAB (Fig. 7b) changes gradually from 55 km BSL in the oceanic crustal domain to 140 km BSL onshore.

## 4   Results

Our results show that the geothermal gradient varies laterally across the models' area and nonlinearly decreases with depth (Fig. 8, 9 and 10; See also Fig. 3 and 4 in the Supplement). To describe these variations, we classified these results in three

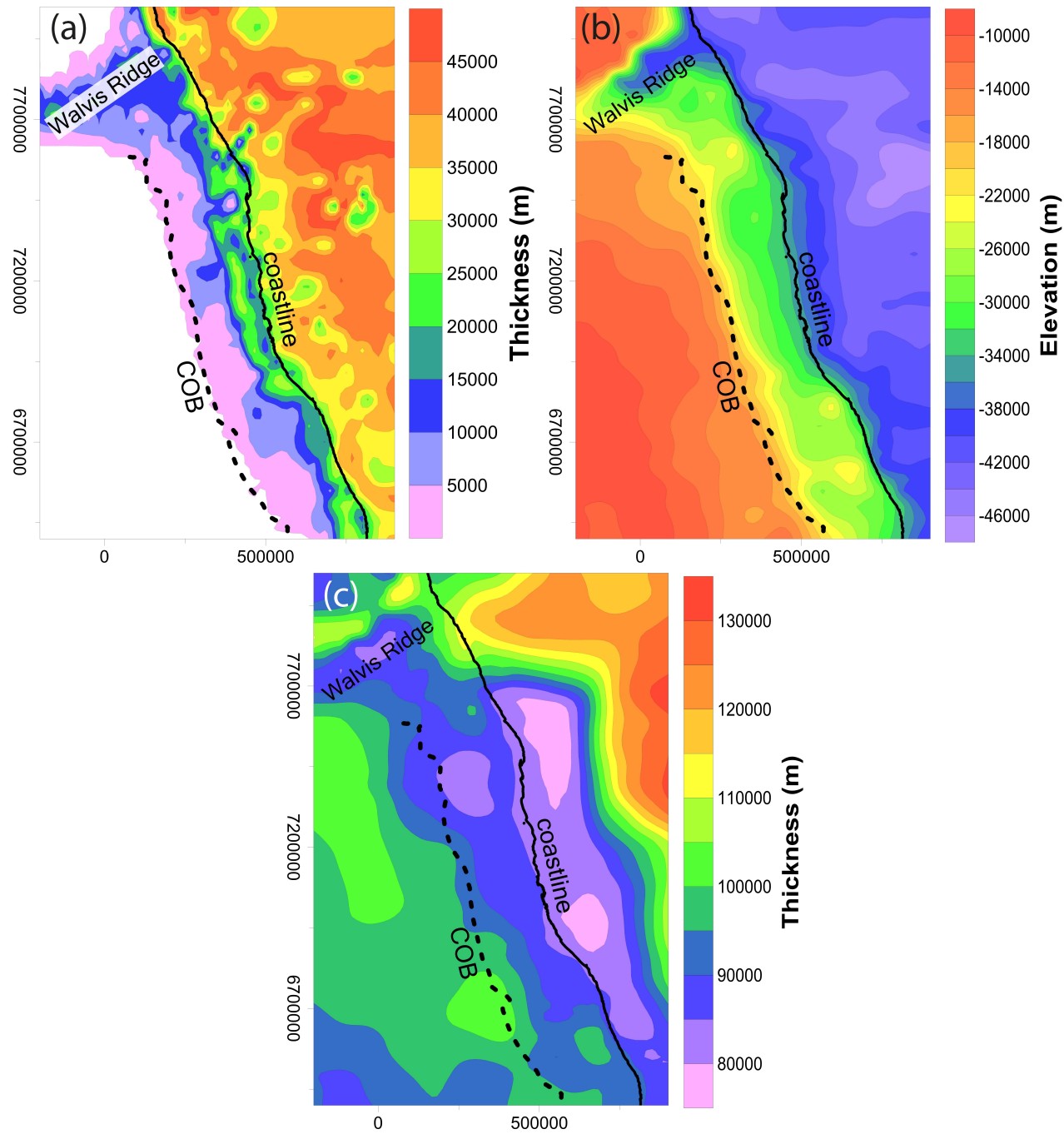

**Figure 3.** 3D structural model of the SW African passive margin: (a) thickness of the upper crystalline crust; (b) depth to Moho; (c) thickness of the lithospheric mantle.

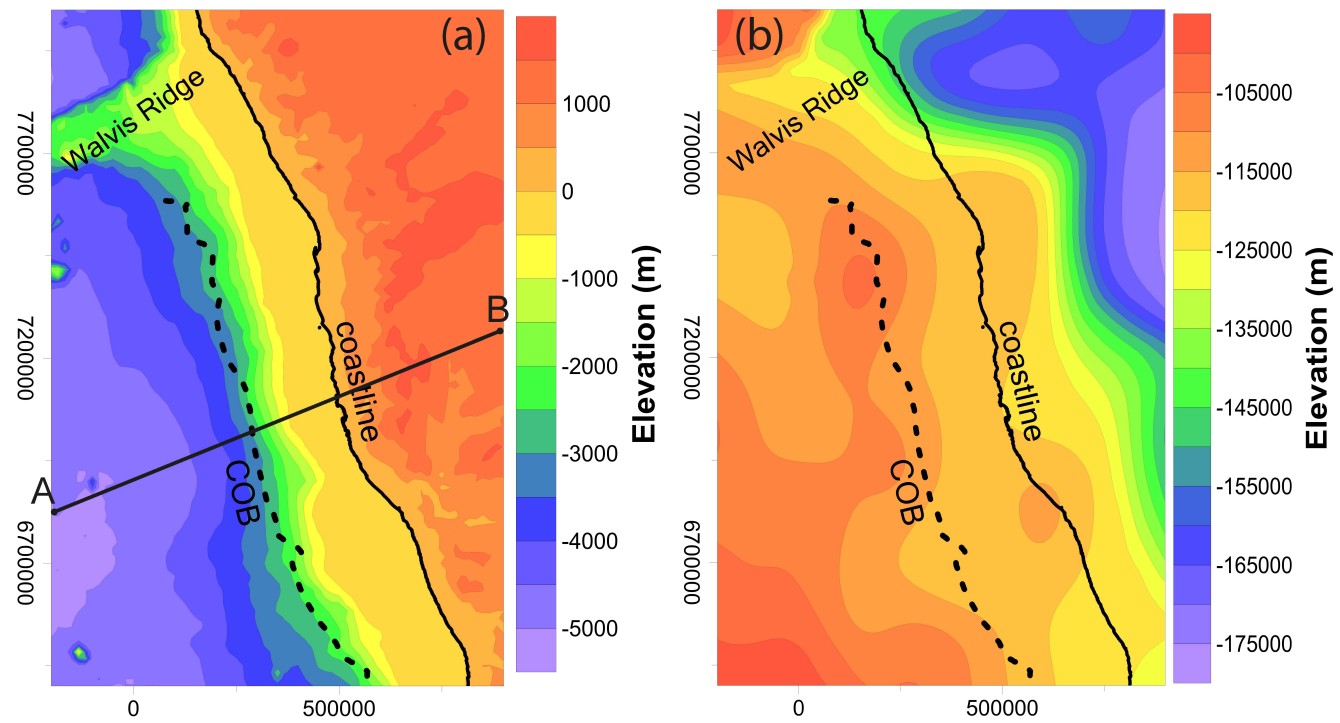

**Figure 4.** Surfaces with a fixed temperature for which the thermal boundary conditions were assigned to in the SW African thermal model: (a) topography–bathymetry corresponding to the upper thermal boundary condition (5 °C); (b) depth of the LAB utilized as the lower thermal boundary condition (1300 °C).

different domains considering the geostructural setting; the onshore domain, the continental margin domain (the area between the coastline and the COB), and the oceanic crustal domain.

## 4.1 The Onshore Domain

In the onshore domain of the SW African model, the geothermal gradient remains in the range of 28–30 °C/km for all depth
5    intervals except for the Precambrian basins (Fig. 8). However, there are some local depressions along the coastline. Within the 2nd (Fig. 8b) and 3rd (Fig. 8c) intervals, the geothermal gradient partly decreases to 26–28 °C/km along the coastline. Within the thicker intervals, this range covers the area more uniformly and to a greater extent (Fig. 8d, 8e, 8f).

Over the onshore domain, the geothermal gradient in the Norwegian model generally stays in the range of 15–17 °C/km for all depth intervals (Fig. 9), and this is the lowest value of the geothermal gradient across the entire model domain. Across
10   the coastline, the geothermal gradient increases steeply seaward from 17 °C/km to 27 °C/km within the first depth interval (Fig. 9a), which is related to the transition between crystalline crust onshore and sediment fill offshore. The same pattern, but with different ranges, is also recognizable for the thicker intervals (Fig. 9).

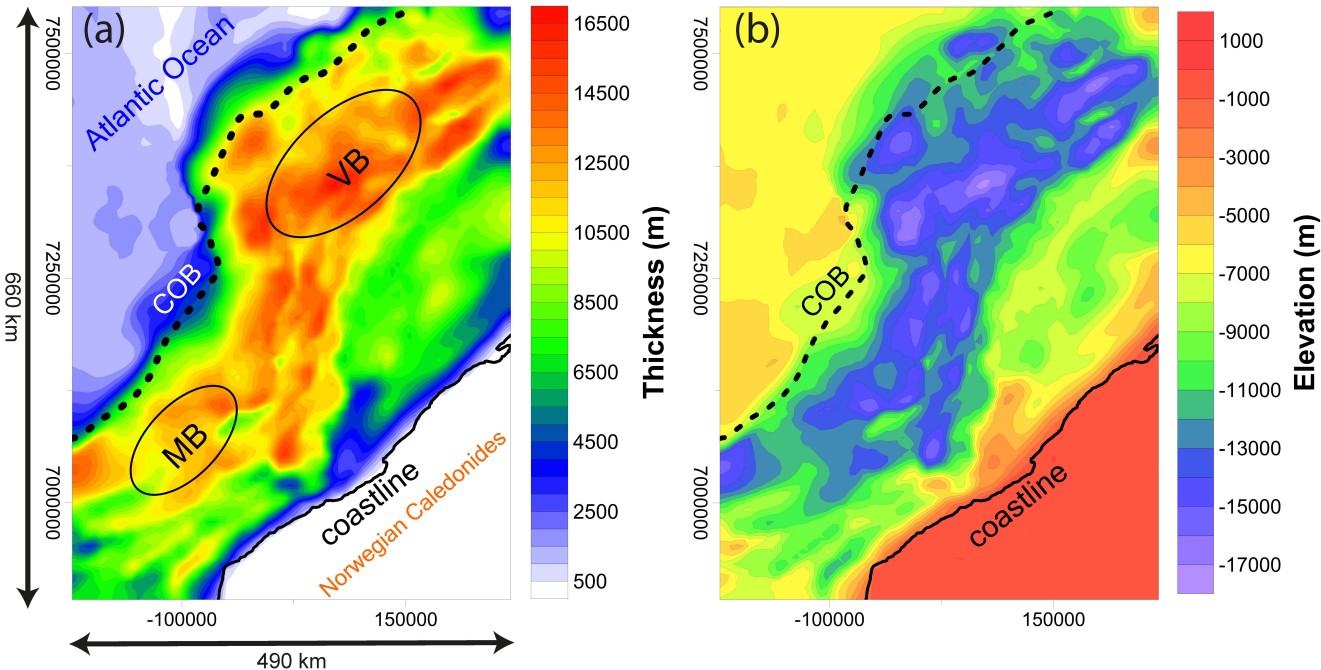

**Figure 5.** 3D structural model of the Norwegian margin: (a) cumulative sediment thickness; (b) depth to top-basement (COB: Continent-Ocean Boundary, VB: Vøring Basin, MB: Møre Basin, UTM: WGS84, zone 33N).

In general, geothermal gradients in the unthinned onshore domain of the SW African margin are greater than in the corresponding domain of the Norwegian margin (Fig. 8 and 9).

## 4.2 The Continental Margin Domain

In this domain, the geothermal gradient variations also reveal a general trend of reduction with increasing depth. Nevertheless,
5   this general trend displays different lateral variations for each sedimentary basin.

### 4.2.1 The SW African Passive Margin

In the SW African model, the results display different patterns of variation for the individual depth intervals. Within the first depth interval (Fig. 8a), the variations are similar in the Walvis and Lüderitz basins. The geothermal gradient increases seaward from the coast and reaches the largest value (48–50 °C/km) in the central parts of the sedimentary basins where the sediments
10   are thickest (Fig. 2a). Oceanward, the gradient declines again towards the distal shelf where the geothermal gradient is in the range of 38–40 °C/km along the COB. In contrast, the geothermal gradient follows a different pattern within the Orange Basin. It decreases with distance from the coast, reaches the lowest value (34–36 °C/km) in the central part of the basin and then increases to the COB. The reduced gradient within the first depth interval in the Orange Basin compared to the two

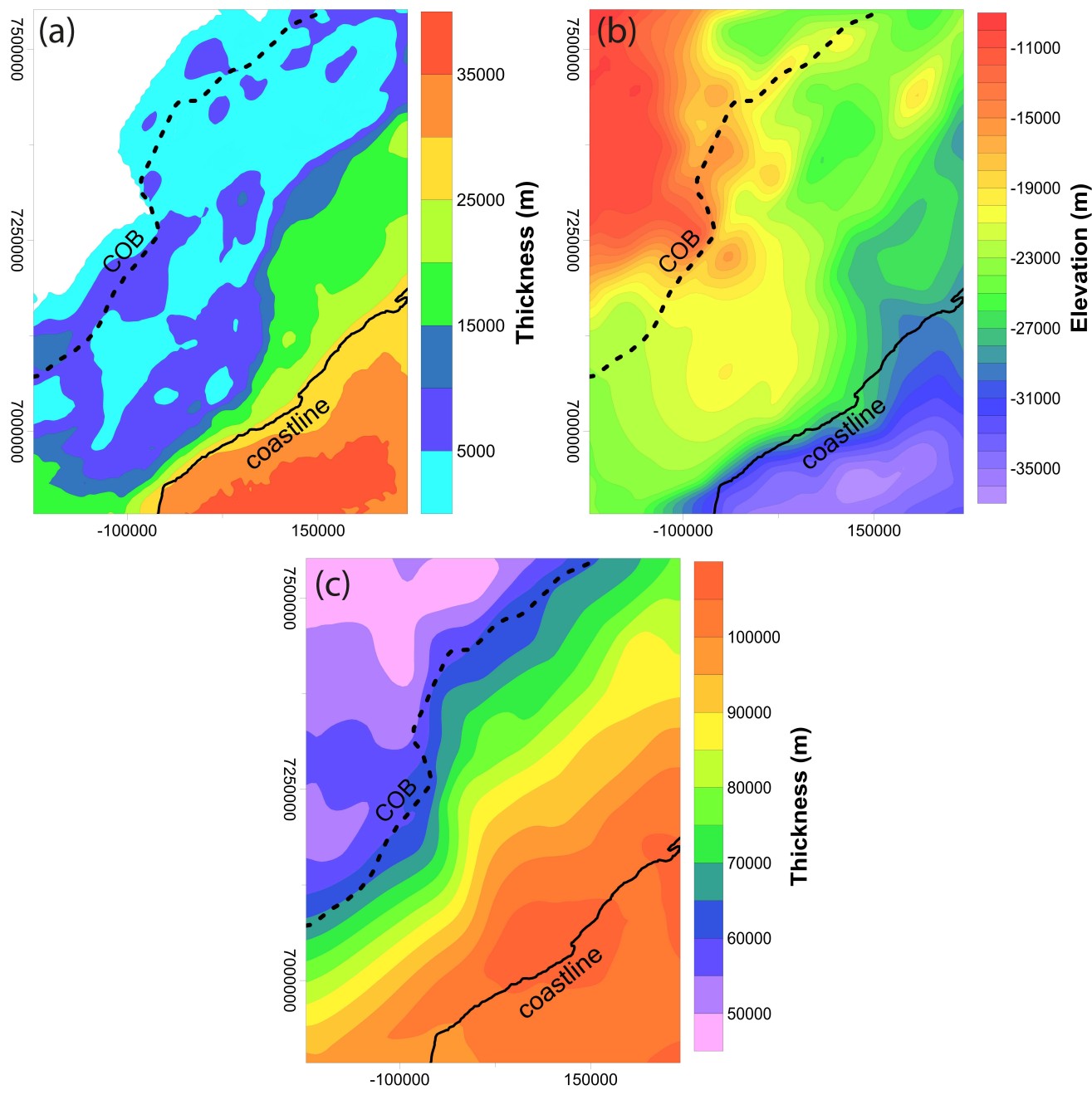

**Figure 6.** 3D structural model of the Norwegian margin: (a) thickness of the upper crystalline crust; (b) depth to Moho; (c) thickness of the lithospheric mantle at the Norwegian continental margin.

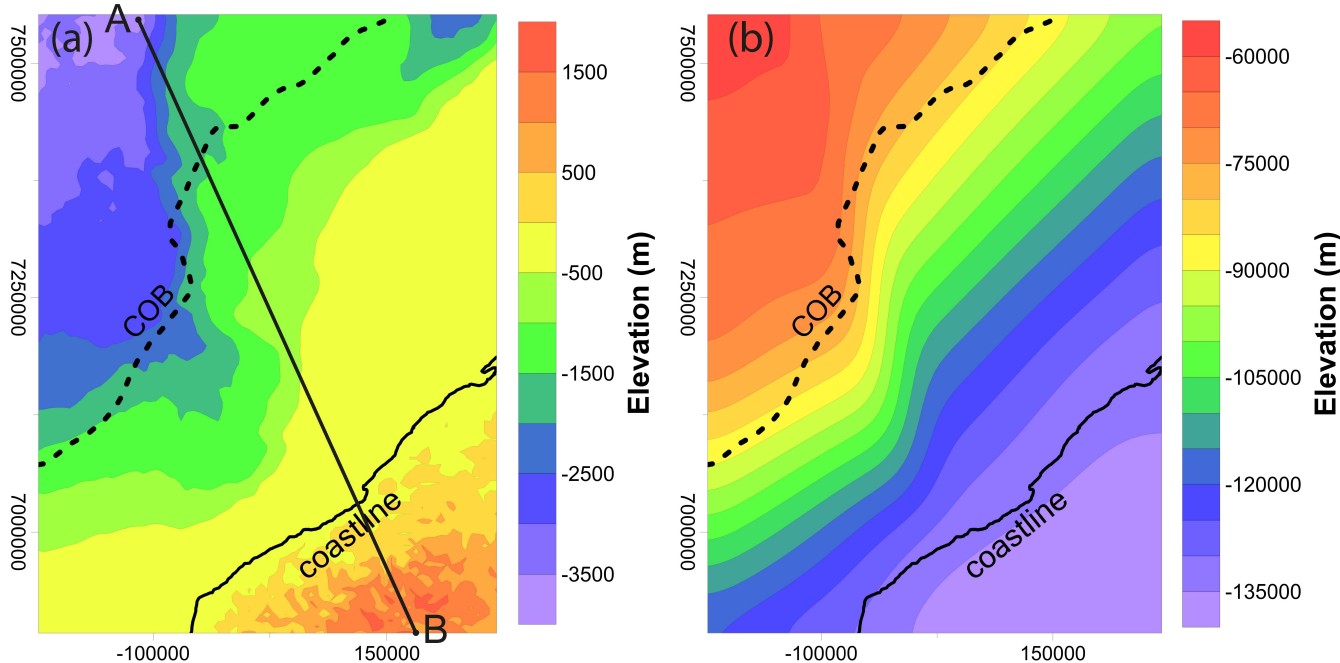

**Figure 7.** Surfaces with a fixed temperature for which the thermal boundary conditions were assigned to in the Norwegian thermal model: (a) topography–bathymetry corresponding to the upper thermal boundary condition (2 °C); (b) depth of the LAB utilized as the lower thermal boundary condition (1300 °C).

other basins corrolates with an increased thickness of the uppermost sedimentary unit of the Cenozoic. This unit has the lowest thermal conductivity of the sedimentary units (Table 1) and is almost absent in the central part of the Orange Basin (Maystrenko et al., 2013).

The variations of the geothermal gradient within the second, the third, and the fourth depth intervals (Fig. 8b, 8c & 8d) follow
5    the same trend as in the first depth interval. A notable difference between these three depth intervals (2nd, 3rd, and 4th) and the first depth interval is the location of the highest geothermal gradient. Within the second, third, and fourth depth intervals, these high values occur in the northern part of the Walvis Basin; this contrasts with the first depth interval where the highest value was found in the Lüderitz Basin (Fig. 8a). These high values are in the ranges of 44–46 °C/km, 42–44 °C/km and 40–42 °C/km within the second, the third, and the fourth depth intervals, respectively. This difference is explicable by considering
10   the top-basement depth (Fig. 2b) and the crustal thickness (Fig. 3a), which are shallower and thicker beneath the northern part of the Walvis Basin compared to the Lüderitz Basin. Moreover, the youngest sediments (with lower thermal conductivity) are thickest in the Walvis Basin (Table 1, Maystrenko et al. (2013)), which is an additional reason for these values of high geothermal gradient within the Walvis Basin.

Within the two thickest depth intervals (5 and 6 km), the results show a different pattern of the geothermal gradient variations within the Orange Basin (Fig. 8e & 8f). Unlike the upper depth intervals the geothermal gradient varies in a similar manner to the Walvis and Lüderitz basins. The geothermal gradient increases seaward from the coast, reaches the locally highest value in the central part of the sedimentary basin, and finally declines towards the COB.

5 Overall, in the SW African model, the highest values of the geothermal gradient for all depth intervals occur within the sedimentary basins (Fig. 8).

### 4.2.2 The Norwegian Margin

The geothermal gradient distribution maps of the Norwegian margin (Fig. 9) also reveal lateral and vertical variations across the sedimentary basins.

10 In the Vøring Basin, the geothermal gradient varies in a similar manner for all depth intervals, except the thickest interval (Fig. 9). Generally, the gradient increases seaward from the coast, decreases in the central part of the basin, and then increases again toward the COB. Similar to the Orange Basin at the SW African margin, the decrease of the gradient is akin to the central part of the Vøring Basin, where the uppermost Cenozoic sedimentary unit with the lowest thermal conductivity (Table 1) is absent (Scheck-Wenderoth et al., 2007). Within the thickest depth interval (Fig. 9f), the geothermal gradient increases gradually from the coast towards the COB and no reduction occurs in the central part of the basin. The highest values of the geothermal gradient, within the Vøring Basin, are found at the distal shelf. These values are in the range of 53–55 °C/km in the first depth interval and attain 35–37 °C/km within the thickest depth interval.

In contrast to the Vøring Basin, the geothermal gradient within the Møre Basin does not follow a comparable pattern in the first two depth intervals. Within the first interval, the geothermal gradient increases gradually and continuously from the coast towards the COB (Fig. 9a). In the second depth interval, the geothermal gradient increases from the coast to the central part of the basin and decreases towards the COB (Fig. 9b). Within the other four thicker depth intervals (Fig. 9c to 9f), the general trend of the geothermal gradient variations is similar to the first depth interval.

### 4.3 The Oceanic Crustal Domain

The oceanic crustal domain refers to the western side of the COB where the crust is mainly oceanic in composition. Herein,
25 the geothermal gradient variations differ significantly between the SW African and the Norwegian margins.

In the SW African model, the results of the calculated geothermal gradient (Fig. 8) for the oceanic crustal domain and within all the depth intervals indicate a lateral oceanward decrease. The geothermal gradient gradually decreases oceanward from the COB to reach the minimum at the western model boundary. These lowest values are in the range of 16–18 °C/km within the thickest depth interval, and 18–20 °C/km within the other five intervals and representing the lowest value of the geothermal
30 gradient over the entire model of the SW African Margin (Fig. 8 and 11c).

In contrast, the results for the Norwegian setting (Fig. 9) show that the geothermal gradient increases oceanward in the oceanic crustal domain, where the highest values of the geothermal gradients over the entire margin are found (Fig. 9 and 11c). From the first depth interval down to the thickest depth interval, these high values stepwise decrease from 53–55, 47–49, 45–47,

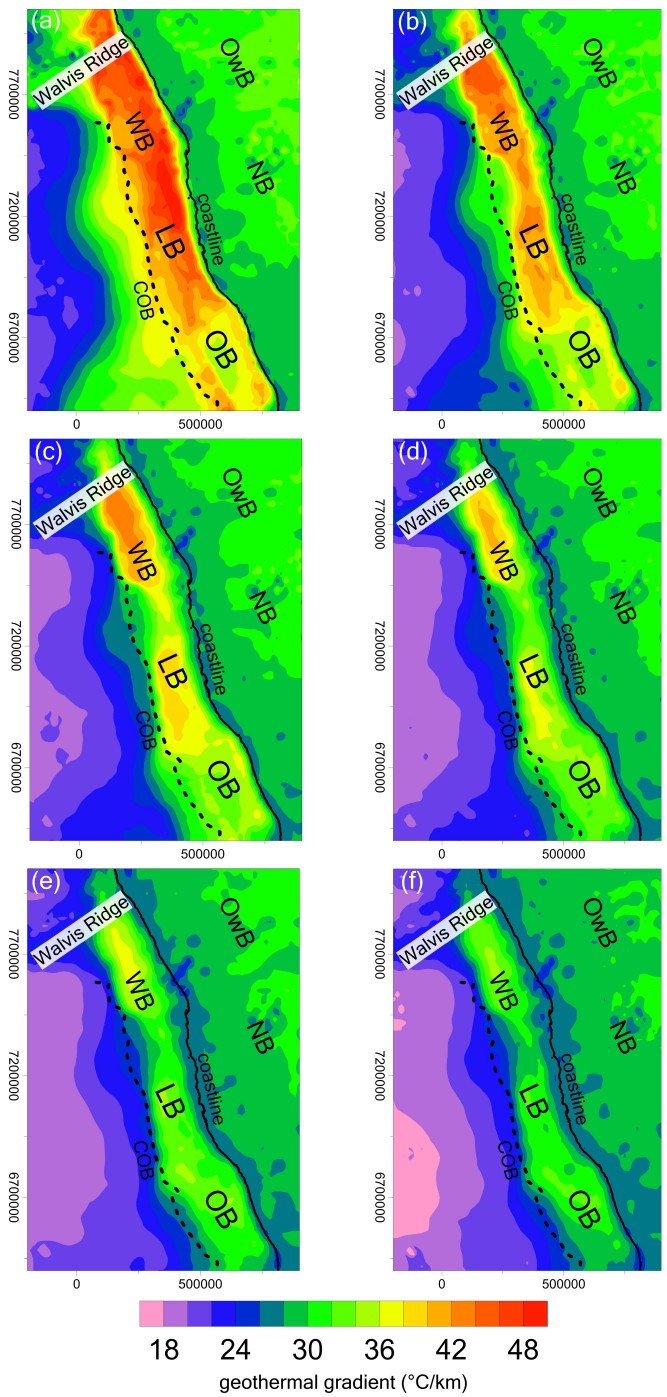

**Figure 8.** Geothermal gradient [°C/km] at SW African margin: the gradient calculated as the temperature differences between the uppermost surface (upper thermal boundary) and the corresponding temperature distribution at (a) 1, (b) 2, (c) 3, (d) 4, (e) 5, and (f) 6 km below the uppermost surface (COB: Continent-Ocean Boundary; Cretaceous-Cenozoic basins: WB: Walvis Basin, LB: Lüderitz Basin, OB: Orange Basin; Precambrian basins: OwB: Owambo Basin, NB: Nama Basin; UTM: WGS84, 33S).

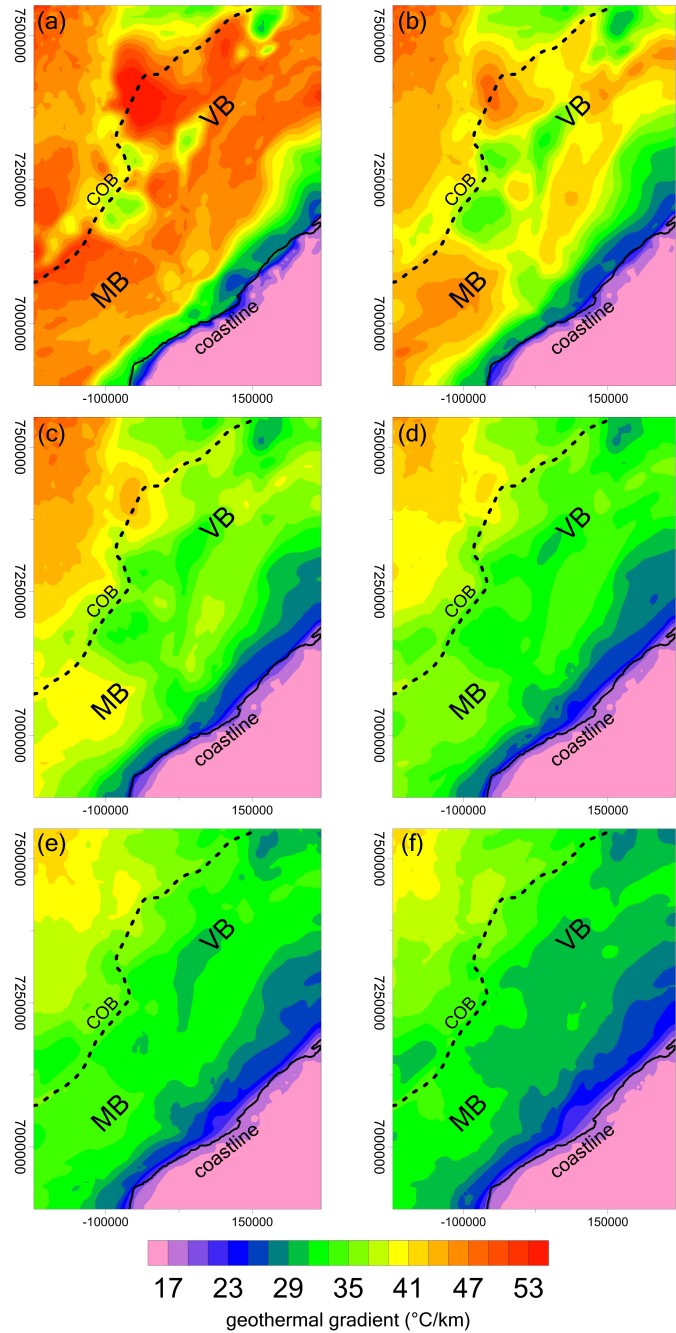

**Figure 9.** Geothermal gradient [°C/km] at Norwegian margin: the gradient calculated as the temperature differences between the uppermost surface (upper thermal boundary) and the corresponding temperature distribution at (a) 1, (b) 2, (c) 3, (d) 4, (e) 5, and (f) 6 km below the uppermost surface (COB: Continent-Ocean Boundary; Cretaceous-Cenozoic basins: VB: Vøring Basin, MB: Møre Basin; UTM: WGS84, 33N).

43–45, 41–43, to 39–41 °C/km. The 65 Ma year difference in the oceanic crust age (SW Africa: 130 Ma; Norwegian: 55 Ma; see Table 1) and the related age-controlled depth of the thermal LAB (Fig. 4b and 7b) would be a reasonable explanation for this difference within the oceanic crustal domain of the two differently aged margins. At the older SW African passive margin the shallowest depth to the LAB is around 100 BSL km below the oceanic crustal domain (Fig. 4b), while the LAB depth at
the younger Norwegian margin is less than 60 km BSL (Fig. 7b).

## 5   Interpretation and Discussion

According to our results, the calculated geothermal gradients reveal variations both laterally and with depth for the two different passive margins (Fig. 8 and 9). In general, the geothermal gradient decreases nonlinearly with depth in both models. However, this occurs in different trends for the two settings. Calculation of the geometric mean value of the geothermal gradient fully
shows nonlinear decreasing from the first depth interval to the thickest depth interval by 40 °C/km to 30 °/km in the Norwegian margin and by more than 32 °C/km to less than 26 °C/km in the SW African margin (Fig. 10). To address the differences between the present-day thermal field of the SW African passive margin and the Norwegian margin, it is important to compare the geothermal gradient variations with the geological structure, the thermal properties of comparing geological units, and the ages of the oceanic crust (Table 1). A structural-thermal cross section (Fig. 11a and b) and corresponding profile of average
geothermal gradient (Fig. 11c) provide supplemental indications for a valid interpretation of shallow thermal field variations across the two differently aged passive margins. We will discuss these issues with regards to the three previously mentioned domains: (1) the onshore domain, (2) the continental margin domain, and (3) the oceanic crustal domain.

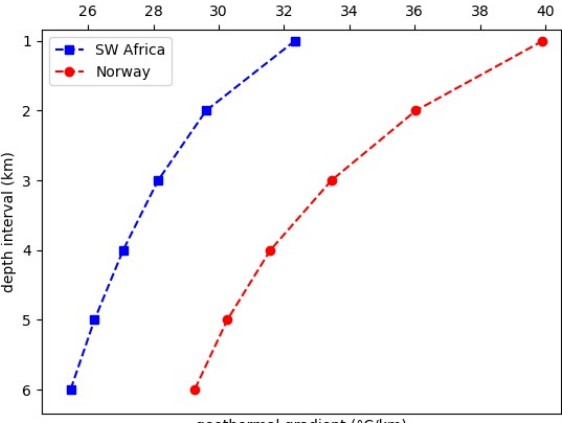

**Figure 10.** The thermal gradient variations with depth: Mean values of the geothermal gradient [°C/km] across the models' area and for the different depth intervals. The temperature-depth distribution maps are presented as figure 1 and 2 in the Supplement.

Temprature-depth function is simply a linear concept of the geothermal gradient (Eq. 3). However, as can be seen from the two models, the geothermal gradient depends on the considered depth interval and varies nonlinearly with depth. The solution to the steady-state thermal diffusion equation (Eq. 4 derived from Eq. 2) is a second-order (nonlinear) temperature function of depth if radiogenic heat production is considered ($S \neq 0$). This fact indicates that the temperature-depth ($T - z$) curvature is highly sensitive to the amount of radiogenic heat production. The interaction of the thermal conductivity of different lithospheric layers and the heat internally produced by the decay of radioactive elements overprint the heat input from larger mantle depth into the lithosphere (Eq. 4). For better comprehension and further comparability, investigating the variability of the geothermal gradient requires representing the same depth intervals across the study areas.

$$\frac{\partial^2 T}{\partial z^2} = \frac{S}{\lambda} \tag{4}$$

## 5.1 The Onshore Domain

In the onshore domain, the geothermal gradient is considerably higher ($\sim 13$ °C/km) at the SW African margin compared to the Norwegian margin. The SW African margin has a thicker crust compared to the Norwegian margin (Fig. 3a, 6a, 11a) and thus relatively more radiogenic heat is contributed by the crust. Additionally, a second reason could be the assigned values of radiogenic heat production in the thermal models (see Eq. 4). In the Norwegian model, Scheck-Wenderoth and Maystrenko (2008), considered an average crustal radiogenic heat production as 0.8 [$\mu W/m^3$] which is much lower than the corresponding value (1.45 [$\mu W/m^3$]) in the SW African thermal model (Table 1). This low value of the geothermal gradient within the onshore domain in the Norwegian model agrees with downhole temperature measurements in the Scandinavian Caledonides that imply an average geothermal gradient of $\sim 17$–$20$ °C/km (e.g., Maystrenko et al., 2015; Lorenz et al., 2015; Pascal, 2015).

Another impressive characteristic of the thermal field of these two passive margins exists in the vicinity of the coastline. Here, the geothermal gradient decreases by about 2 °C/km at the SW African margin. This reduction spatially correlates with the crustal thickness decrease ($\sim 10$ km) beneath the coast (Fig. 3a). The thinner crust produces less radiogenic heat, which leads to lower temperatures. In contrast, considering the same area at the Norwegian margin, the geothermal gradient increases by approximately 10 °C/km within the first depth interval and gradually decreases within the deeper depth intervals. These variations might be explained by the thermal blanketing effect of the up to 1.5 km thick insulating sediments (low thermal conductivity, Table 1) along the coast (Fig. 5a). While the outcropping crystalline crust onshore efficiently transports heat to the surface in response to its greater thermal conductivity, the heat is stored in the insulating sediments offshore.

## 5.2 The Continental Margin Domain

To interpret the thermal field variations within the sedimentary basins, and to compare these differences between the SW African and the Norwegian margins, we need to take a closer look at the geometry of the geological structural units within and beneath the location of the sedimentary basins. These units were presented in section 3 and here we will discuss how they affect the thermal field.

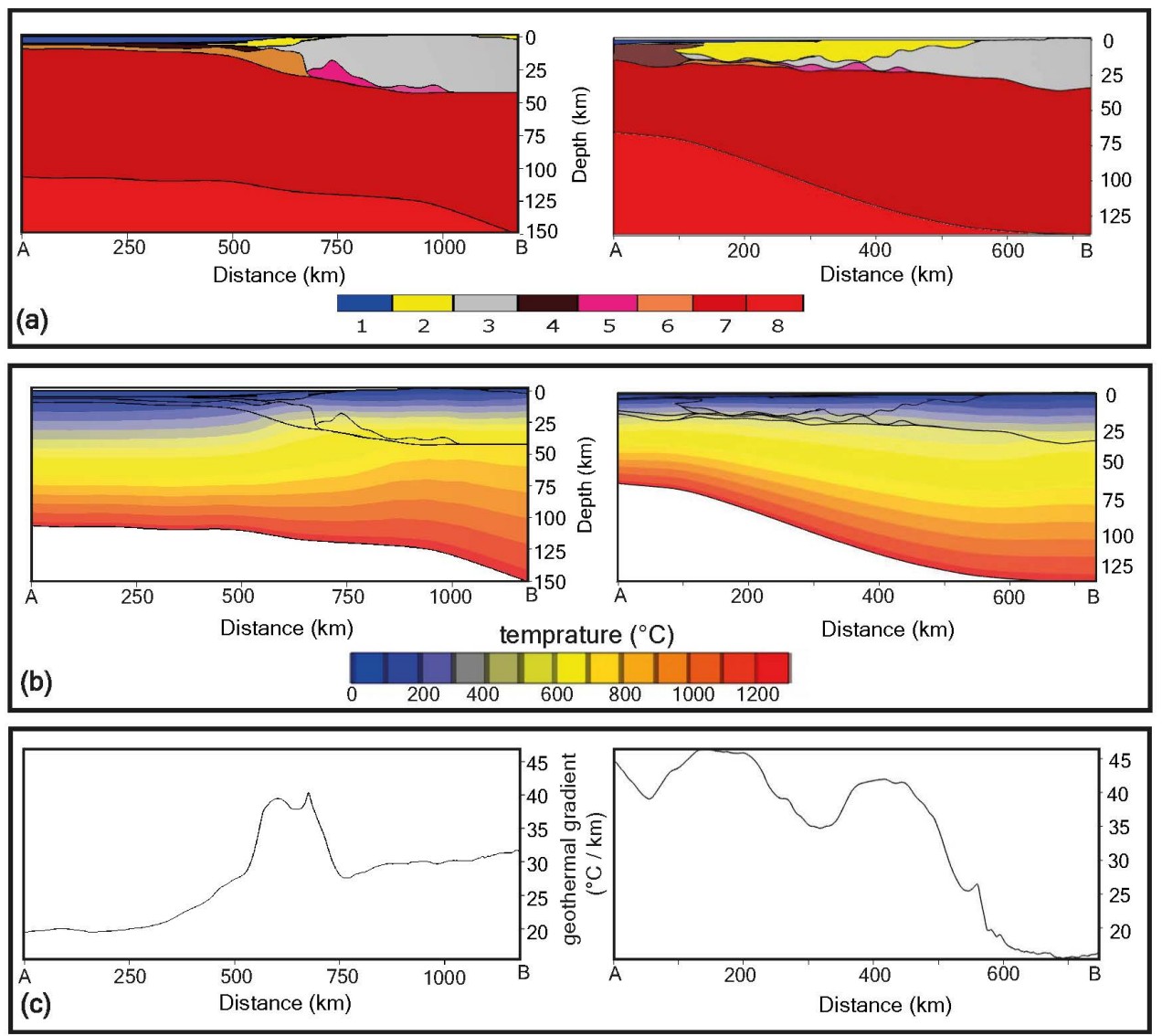

**Figure 11.** Structural-thermal cross section and corresponding average geothermal gradient at SW African passive margin (left) and Norwegian continental margin (right) after Scheck-Wenderoth et al. (2007); Scheck-Wenderoth and Maystrenko (2008); Maystrenko et al. (2013): (a) Structural crustal cross section along A–B profile (Fig. 4a and Fig. 7a). Numbers near color legend for individual layers: 1) water, 2) sediments, 3) crystalline continental crust, 4) oceanic crust, 5) high-density continental crust, 6) high-velocity high-density lower crustal layer, 7) low density mantle, 8) normal density mantle. (b) Temperature distrbution within the A–B cross section. (c) Calculated average geothermal gradient along the A–B cross section.

Heat is transferred from the oceanic domain to the adjacent distal margin as a natural consequence of the 3D heat transport. Thereby the local structural configuration and the related variable distribution of thermal rock properties leads to very specific thermal pattern, be it in the area of "normal" or transfer segments of the margin. Accordingly, our results are consistent with earlier studies analyzing the transition from the oceanic to the continental domain (Nemcok et al., 2012; Henk and Nemcok, 2016).

### 5.2.1 The SW African Passive Margin

Considering the geothermal gradient variations over the whole study area at the SW African passive margin, the highest values for the geothermal gradient occur within the sedimentary basin areas. Beneath the continental margin, the crystalline crust is thinner (i.e. less radiogenic heat production) in comparison to the onshore domain (Fig. 3a). Moreover, the LAB is also deeper beneath the sedimentary basins compared to the LAB depth below the oceanic crustal domain (Fig. 4b). Accordingly, in spite of a lower radiogenic heat production and a larger depth to the thermal LAB, the gradients are highest in the sedimentary part. This indicates that the thermal blanketing effect of the insulating sediments has the strongest control on the shallow thermal field variations within the sedimentary basins, and geothermal gradients widely correlate positively with sediment thickness.

The top-basement (Fig. 2b) is much deeper below the Orange Basin and the radiogenic crust thinner compared to the Walvis and Lüderitz basins ($\sim$ 10 km difference in the center of the sedimentary basins). Accordingly, the thicker sediments within the Orange Basin (Fig. 2a) lead to a more pronounced thermal blanketing effect due to the low thermal conductivity of these sediments. Additionally, Cenozoic sediments with lower thermal conductivity are thicker in the Walvis and Lüderitz basins compared with the Orange Basin (Table 1, Maystrenko et al. (2013)). These differences in the top-basement depth and the thickness of younger sediments with low thermal conductivity would explain why the geothermal gradient has the lowest local value in the central part of the Orange Basin in the upper depth intervals, 1 to 4 km below the upper thermal boundary condition. Within the thicker depth intervals, between 4 and 6 km below the upper thermal boundary condition, all sediments have a Cretaceous age (Table 1, Maystrenko et al. (2013)). Consequently, the thermal field pattern shows more similarity within all the three sedimentary basins and the geothermal gradient increases toward their central part (Fig. 8e & 8f).

### 5.2.2 The Norwegian Margin

In general, the geothermal gradient variations within the sedimentary basins show fewer complexities at the Norwegian margin in comparison to the SW African passive margin. Within the Vøring Basin and for all depth intervals, the geothermal gradient generally increases seaward, decreases in the central part of the basins, but increases again toward the distal shelf (Fig. 9). While the crystalline crust is thinner (i.e. less radiogenic heat is produced) beneath the sedimentary basins compared to the onshore domain (Fig. 6a), the lithospheric mantle (Fig. 6c) gradually thins and the LAB (Fig. 7b) becomes progressively shallower towards the ocean. In addition, the lack of the post-breakup (uppermost Cenozoic) sedimentary unit in the central part of the Vøring Basin reduces the thermal blanketing effect of insulating sediments (Table 1, Scheck-Wenderoth et al. (2007)). With increasing depth, the thermal blanketing effect of Cenozoic sediments gets less relevant for the thermal field variations, while the depth to the LAB plays a more prominent role. As shown in Fig. 9f the geothermal gradient reveals no depression in

the central part of the Vøring Basin, but increases seaward due to the shallower LAB and the thinner lithospheric mantle. The same reason would explain the geothermal gradient pattern that characterizes the More Basin. Overall, the pattern of the shallow thermal field looks similar for all depth intervals in the More Basin, increasing gradually from the continental shelf towards the distal shelf. This trend agrees with the oceanward shallowing LAB depth.

The absolute values of the geothermal gradient within the sedimentary basins in the Norwegian model are larger compared to the corresponding values in the SW African model. The highest geothermal gradient at the SW African passive margin occurs within the sedimentary basins (Sec. 5.2.1), whereas this is not the case for the Norwegian model. In addition to the quantitative differences, these high values exist within the central part of the sedimentary basins at the SW African margin, while for the Norwegian model the highest local values of the geothermal gradient in the continental margin domain occur closer to the

distal shelf. These differences indicate fundamentally different controlling factors for the shallow thermal field at these two differently aged passive margins.

## 5.3    The Oceanic Crustal Domain

The oceanic crustal domain is most important for comparing the shallow thermal field variations for the SW African passive margin and the Norwegian margin. While the SW African model has the lowest values of the geothermal gradient in this

domain (Fig. 8), the Norwegian model presents the highest value of the geothermal gradient (Fig. 9).

The volcanic passive margin of Norway (55 Ma) is significantly younger than the SW African passive margin (130 Ma). This age contrast resulted in around 40 km depth difference of the thermal LAB for these two passive margins (Fig. 4b & Fig. 7b). The consequence of this shallower oceanic LAB is a steeper average geothermal gradient as the 1300 °C difference between surface and LAB needs to be accommodated within 60 km. This distance is almost twice as large at the SW African margin

where the oceanic LAB is at 110 km depth. Accordingly, the young Norwegian margin is hotter in comparison to the old SW African margin, which appears to be thermally equilibrated (Maystrenko et al., 2013).

One clear point stands out and that is the first order difference in the age of breakup at the two margins. Also, the timing of potential interaction in the oceanic opening with a mantle plume is different. For the South Atlantic, recent results (Scheck-Wenderoth et al., 2017) suggest that the breakup was not initiated by a mantle plume, but some interactions with the Tristan

da Cunha hot spot may have influenced the evolution. Both the opening and the potential breakup-plume interactions were terminated at about 130 Million years b.p. Thus assuming steady-state thermal conditions today is a valid hypothesis as thermal equilibration has been achieved. This is also expressed in the thick oceanic mantle lithosphere derived from seismological and gravity data.

In contrast, the Iceland plume is next door to the Norwegian North Atlantic margin today (Steinberger and Torsvik, 2012) and

breakup is significantly younger (55 Million years b.p.). Given the time-thermal constant of the lithosphere thermal disturbances older than 65 Ma would largely have equilibrated (Turcotte and Schubert, 2014). This implies that the thermal consequences of breakup should have declined after 55 Million years and that the system is at least close to thermal equilibrium today. Though this consideration does not account for additional dynamic thermal effects induced by the Iceland plume we use a steady-state thermal model (Scheck-Wenderoth and Maystrenko, 2008) to assess the deep thermal field for two reasons: (1) We aim to base

our comparison of the two margins on the same method and (2) Both models are consistent with observed temperatures in the upper few km (e.g., Channell et al., 2006) and with surface heat flow (e.g., Ritter et al., 2004). We would additionally like to point out that the potential error related to the steady-state assumption would, in the worst case, cause an underestimation of the mantle heat input, but not the other way round. To reach the high observed shallow temperatures with a transient simulation, while respecting at the same time the thermal lithosphere base indicated by seismology and gravity, would either require a higher temperature at the base of the lithosphere (if the process is at an early stage) or a late stage in thermal equilibration (close to steady-state).

A further discussion point relates to the nature of the lower crustal body. Depending on which lithology is assumed for the latter, higher (mafic underplate: gabbro) or lower (serpentinized mantle) thermal conductivities and different amounts of radiogenic heat production would result. Autin et al. (2016) have examined the thermal implications of these different compositions for lower crustal bodies at the Argentine magma-rich margin of the South Atlantic. They found that apart from the serpentinite model being colder, the thermal effects of gabbro and eclogites would be similar.

Proximity to Iceland mantle plume might also be effective in causing high geothermal gradients in the oceanic crustal domain of the Norwegian margin. The North Atlantic breakup was possibly initiated by the abnormally hot mantle of the Iceland plume (White, 1989; Skogseid et al., 1992; Gernigon et al., 2004, 2006; Parkin and White, 2008) activated approximately 5 million years earlier than the continental breakup (Saunders et al., 1997). While some studies have shown that the Iceland plume propagated northward (e.g., Ruedas et al., 2007; Steinberger et al., 2015), seismic tomography (Rickers et al., 2013) suggests lateral movement of plume material in addition to the parallel propagation along the mid ocean ridge. Moreover, 3D thermomechanical models (Koptev et al., 2017) suggest that plume-related thermal perturbations such as hot mantle lateral flows may result in topography at the Norwegian passive margin with long wavelength variations onshore and short wavelength variations offshore. To quantify such effects, future studies need to consider observation-based configurations, but also implement the physics of processes controlling mass and energy transport related to mantle flow dynamics.

With respect to the hypothesis formulated in the introduction that the present-day thermal field at passive continental margins is determined by lithospheric mantle characteristics, our results suggest that considering variations in the crust alone is not sufficient to properly assess the present or past thermal configuration of passive continental margins.

## 5.4 Implications

More recent studies apply complex modelling approaches to simulate thermal histories considering spatial and temporal variations in temperature profiles (e.g., Person and Garven, 1992; Bertotti and Ter Voorde, 1994; Ehlers and Farley, 2003; Ehlers et al., 2003). However, these methods of thermal history reconstruction are mostly based on paleotemperature indicators, that experience irreversible structural changes when passing through a certain temperature window (Allen and Allen, 2005; Naeser and McCulloh, 2012), but do not consider observation based on three-dimensional structural settings. The thermal alteration of organic matter for example results in specific changes of vitrinite reflectivity and linear relationships between temperature and vitrinite reflectivity have been established using lab experiments (Dow, 1977; Barker and Pawlewicz, 1986; Burnham and Sweeney, 1989; Corcoran and Clayton, 2001). Likewise, Apatite Fission Track Analysis makes use of the specific temperature-

dependent behaviour of fission track in response to radiogenic decay (Barker, 1996; Gallagher et al., 1998; Stockli et al., 2000; Reiners and Brandon, 2006; Deeken et al., 2006). Such paleotemperature indicators are often translated to amounts of paleo-burial depth assuming a constant paleo-thermal gradient for a certain study area and the difference between the present-day depth and the paleo-depth is interpreted in terms of vertical movements. Our results indicate that the thermal gradient may vary significantly both laterally and with time (Fig. 11). Accordingly assuming an average paleo-thermal gradient of 30 °C/km positions the 70°C window of an Apatite sample at 2 to 3 km depth whereas a higher paleo-geothermal gradient of 45 °C/km would position the same sample at 1.5 km depth. Therefore, considering paleo-geothermal gradient variation in response to sedimentation or lithosphere cooling is key if paleo-temperatures, paleo-elevations and, derived from the latter, vertical movements are deduced. This implies that in addition to the general paleotectonic setting, also the evolutionary phase and the position in this setting need to be considered. For passive margin settings this means that it is not only important to take into account the type of passive margin (magma-rich versus magma-poor) but also the location (relative to the continent and to the newly formed oceanic domain) and the time with respect to break up are relevant to consider. A sample from a proximal or distal domain at an early or late stage of evolution has experienced different thermal imprints and the paleo-position should be considered accordingly in thermal history reconstruction.

# 6 Conclusions

The assessment of variations in the geothermal gradient for the two different passive volcanic margins revealed that:

- In spite of a similar crustal structure, the geothermal gradient differs laterally across the two passive margins and non-linearly decreases with depth.

- The thermal field of the two margins is contrasting. At the Norwegian margin (young) the thermal field is mostly dominated by the thermo-tectonic age and the thermal LAB depth in contrast to the SW African margin (old) where the crustal configuration is dominating the pattern of the equilibrated shallow thermal field.

- Over the onshore domain, the radiogenic heat production is the main heat controlling factor for both settings. Within the sedimentary basins, the thermal blanketing effect of the insulating sediments has the highest impact on the shallow thermal field at both margins. In the oceanic crustal domain, the thermal field is highly affected by the age of the ocean and the thermal LAB depth. Therefore, the Norwegian model is significantly hotter than the SW African model in the oceanic crustal domain and in the distal margin.

- While the causative thermal anomaly leading to margin formation in the South Atlantic should be equilibrated, the thermal disturbance in the North Atlantic and the proximity to the Iceland plume obviously cause thermal effects at present-day. Characteristics of the lithosphere ultimately determine the thermal field for the two settings.

- This fact that the geothermal gradient is nonlinear and varies across areas has implications for methods of thermal history reconstruction.

*Competing interests.* The authors declare that they have no conflict of interest.

*Acknowledgements.* The research leading to these results has received funding from the People Programme (Marie Curie Actions) of the European Union's Seventh Framework Programme FP7/2007-2013/ under REA grant agreement n° [607996]. We are very grateful to Yuriy P. Maystrenko for his feedback on the database, and to Jessica Freymark for her methodological support. All data for this paper is properly cited and referred to in the reference list.

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
