# Peer review of "Variability of the geothermal gradient across two differently aged magma-rich continental rifted margins of the Atlantic Ocean: The Southwest African and the Norwegian margins"

_Solid Earth, 2017_

## Referee Comment (RC1) · Anonymous Referee #1 · 5 Oct 2017

General comments

In this paper "Variability of geothermal gradient across two differently aged continental volcanic passive margins: The Southwest African and the Norwegian margins", Gholamrezaie et al. present the results of two 3D models of the conductive thermal state in magma-rich rifted margins. They propose that the geothermal gradient is highly variable in time and from one margin to another: (1) in the amplitude of the geothermal gradients, (2) in the lateral distribution across the margin, (3) in the sedimentary basins. One of the main controlling parameter is the age of the breakup and thus the age of

the oceanic lithosphere.

The authors stress that this evolution of the geothermal gradient with time, its non-linearity with depth and its spatial variability has major implications for the calculation of paleo-temperatures and paleo-elevations (as very simple and constant paleo-gradient are used).

I think the manuscript can be improved by clarifying some points and by some additional discussions, all of which is discussed in detail below. In particular, the figure captions could be improved and more recent references could be added. In summary, this paper describes topics and an area of interest to a large audience and will be a very nice contribution to Solid Earth Discussion after minor revision.

Specific comments

I would suggest adding recent references in the geological settings of the two margins. Both area are widely covered by a substantial number of publications and only 4 references are more recent than 2010.

The authors state that the Norway margin is not in an equilibrated thermal state. How does this result agree with the calculation of a steady-state (i.e. equilibrated) conductive thermal model? Maybe this could be discussed.

There is no reference for the statement: "One of the typical characteristics of volcanic passive margins is an extremely thinned continental crust ($\sim$ 5 km in the distal margin)" (p. 5 line 9). Actually, I would rather state that it is a characteristic of magma-poor rifted margins. Hyper-extended continental crust in magma-rich settings is proposed, but with consequent magmatic additions that increase the thickness of the crust.

Figure 10 illustrates the evolution of the mean thermal gradient with increasing depth intervals. Maybe it could be helpful for the reader to represent the more classical "temperature vs depth" graph for each margin. It would be interesting to represent it also for each domain. While it is not well constrained, the author could also propose

an evolution of this gradient through time in order to better illustrate the time variability.

In the conclusions, you mention the influence of the Iceland plume. However, the role of plumes and their relative timing with the breakup (wide debate) are not discussed at all in the paper. A brief paragraph could be added.

Technical corrections, minor suggestions

Title :The margin community would rather use "magma-rich rifted margins" than "volcanic passive margins" but this is not a problem for the understanding of the paper.

Part 2.2: I did not understand the significance of "upper thermal boundary" at first. Maybe it should be defined.

Part 2.2: The order of citation of the figures in the text is not respected: figure 1 and then directly figure 4.

Part 2.2, line 26: Please explain why it is more relevant to use increasing thicker intervals of calculation with depth.

Part 4.3, line 28: please remind the reader the ages.

Part 5, line 3: ass an "s" to "margin".

Part 5.3, line 11: Please remind the reader all the mentioned hypothesis.

Part 6, line 12 "in" instead of "of".

All maps: a short title on each map would greatly help the reader.

Figure 2: Does the sediment thickness map includes Proterozoic sediments?

Figure 8 and 9: Abbreviations should be explained in the captions. Precambrian basins could be highlighted. The color scale changes for each map: would it be interesting to keep it the same for each interval?

Figure 10: It would be more logical to attribute the blue color to SW Africa and the red

color to Norway as it is hotter.

---

## Referee Comment (RC2) · Anonymous Referee #2 · 2 Nov 2017

**Review of the article Variability of geothermal gradient across two differently aged continental volcanic passive margins: The Southwest African and the Norwegian margins by Gholamrezaie et al.**

This paper represents an interesting addition to the research focused on cooling histories of continental margins and their controlling factors. Although I have enjoyed reading it, it can be further improved to reach good quality. Below are my key points suggesting revisions of specific items. Given the extent of suggested revision, it can be classified as moderate.

General comment:

Paper needs to demonstrate more awareness of previous work. While the previous work on specific chosen margin examples is cited a bit better, the paper seems to work in relative isolation from studies done on thermal histories and their controlling factors.

Specific items:

**Abstract:**

Line 11 – make a full term and place LAB into parentheses. Otherwise your reader has to wait until she gets to page 7 to understand what you mean.

**Text:**

*Introduction:*

p. 1, lines 16 and 21: Order your citations according to the publication year. Do this in the entire manuscript.

p. 1, line 21: Use original references instead of relatively modern ones wherever applicable to honor the scholars who came up with certain idea originally. Do this in the entire manuscript.

p. 2: When you are introducing the thermal history of both the oceanic crust and lithospheric mantle, you need to use the knowledge from the pioneers of this research: Parsons and Sclater (1977), Stein and Stein (1992), Goodwillie and Watts (1993), Watts and Zhong (2000)

***Method:***

p. 3: Tab. 1 needs a bit of explanation either in figure caption or, preferably, in the text. The reason for this is that you are making a claim that both margins have a very similar configuration of crust (p. 2, lines 29-30) but Tab. 1 indicates a large difference in characterizing the average thermal conductivities of oceanic crust, and a distinct difference between the conductivities of high-velocity bodies. Such a difference should have an impact on the thermal history of these two study areas.

p. 4, line 24: Make sure that your figures are cited in ascending order in your text. Here you are making a jump from Fig. 1 directly to Figs. 4 and 7.

p. 4, lines 25-27: It would be better to compare "apples and apples". Instead of comparing thermal gradients of 1 km-thick layers, you are comparing those of layers which are progressively 1km thicker than each previous one. I know that you can still see the downward-decreasing gradient using this approach, but aren't we supposed to compare the most directly comparable things when we do the research based on a comparison?

p. 5, Fig. 1 caption: Here you wrote a caption, which could make an impression that you are calculating thermal gradients for six 1km-thick layers. Make sure that this caption is in accordance with your text on p. 4, lines 25-27.

***Exploited models:***

p. 5,  lines 9 and also 11: You can use older original references, rather than a random choice of younger ones. It would show your command of literature and understanding, which studies brought the original knowledge and which ones were just developing it further. Suggestions: use some of the articles by Huismans and Beaumont and some of the articles with Manatschal co-authoring, for example

p. 6, lines 3 and 10-12: Use just 2-3 references for specific knowledge item. You will save some space. Make sure that you choose the original ones for the idea.

p. 6, line 16: Here you are describing a similar character of both margins. However, this could be a good place to discuss those differences in thermal conductivities from Tab. 1 to lay down the groundwork for your later discussion about reasons for 2 different thermal histories. Here you can also touch on different thickness and distribution of sedimentary cover …etc. Honestly, when I look at your Fig. 11 a, the two margins look rather different. Thicknesses of adjacent oceanic crusts are very different. Thicknesses and geometries of sediments – very different, volumes and geometries of high-velocity bodies – very different, geometry of thinning in the crystalline continental crust – very different. If you do not make

a thorough comparison, your reader may think that you have found very similar margins where one can see what the different time for the dissipation of rifting/breakup-controlled transient does to their present-day thermal regime. However, your case requires much more thinking involved in the comparison of the two margins because the ratio of various interacting factors in control of their present-day thermal structure is different.

p. 6, line 26: "… COB after Pawlowski (2008)…" should be rather described as "COB determined from ….this and this constraining data (Pawlowski, 2008)…"

p. 10: You also have one more problem to discuss, if you want to compare the two chosen margin examples, because they are not "apples and apples". While the S Atlantic one is a pure rifted margin, the Norwegian one has a large transform margin segment dividing the Møre and Vøring rifted margin segments. The two Norwegian margins are also characterized by being tectonically and thermally affected by multiple rifting events, instead of a single one that evolved into the breakup in S Atlantic case. How do you filter out these two effects in the case of Norwegian example to be able to compare the two case margins with respect to their controlling factors such as the LAB geometry, thermal blanketing by young sediments and thinning geometry of the continental crust?

***Results:***

p. 13, line 18: replace "theses" by "these"

p. 14 and 15, Fig. 8 and 9 captions: This caption describes already a third version of your thermal gradient calculation, now letting your reader think that they are calculated at a set of six depth levels, the deepest one being 6 km deep. Make sure that your manuscript carries a unified story of your thermal gradient calculation and display.

p. 16: The Norwegian Margin: Here you need to do more than the descriptions of geological reasons for gradient distributions that you have here. The reason is that when you want to compare various geological reasons for such complex (and not equilibrated yet) Møre-Vøring thermal field, you need to know that:

It is the deformation history that has a controlling role on the tectonic and thermal development, as concluded from a comparison of Møre and Vøring neighbor margins in Norway (Fernandez et al., 2005). The differences of the magma-rich Vøring margin from magma-poor Møre margin are:

1)      the occurrence of the extra rifting event at the beginning of the rifting history;

2)      two times thicker underplated body underneath the distal margin;

3)      30 km thicker original Caledonide lithosphere;

4)        a slightly smaller stretching factor;

5)        larger thickness of adjacent oceanic crust; and

6)        a 10 km thinner lithosphere underneath the distal margin.

These differences were attributed to different rifting histories, including the enhanced heat transfer from the oceanic crust adjacent to the Møre margin to continental crust of the Vøring margin through the contact provided by transform and occurrence of the ridge jump responsible for the separation of the Jan Mayen micro-continent initially adjacent Møre margin (Fernandez et al., 2005).

The cumulative length of rifting events at a magma-rich Vøring margin is long. The extension initiated here in late Permian and ended by Paleocene/Eocene break-up, comprising late Permian-Triassic, late Jurassic-Early Cretaceous, Late Cretaceous-Paleocene extensional events (Ziegler, 1989; Brekke, 2000; Skogseid et al., 2000; Gernigon, 2002; Van Wijk and Cloetingh, 2002).

The regional crustal stretching and subsequent crustal necking in the Vøring scenario is characterized by the last activity timing shift towards the stretching axis (Geoffroy, 1994, 2005; Schlindwein and Jokat, 1999). Unlike the S Atlantic example, in the Vøring example the crustal stretching and stretching/necking transition took about 204 Ma, although characterized by discontinuous extension. The extension initiated in late Permian in outboard locations and continued until the Paleocene/Eocene boundary in inboard locations (Ziegler, 1989; Brekke, 2000; Skogseid et al., 2000). The Paleocene was characterized by the emplacement of traps that buried the pre-existing Late Cretaceous normal fault patterns (Geoffroy, 1994; Gernigon, 2002). The transition from crustal stretching to necking then took place rather quickly, during Paleocene/Eocene transition, culminating with the break-up (Gernigon, 2002; Van Wijk and Cloetingh, 2002). While the Mesozoic stretching rate was as low as $7*10\text{-}16s^{-1}$, taking place during 75 Ma (Gernigon, 2002), the Paleocene/Eocene stretching-necking transition was exceptionally fast (Hinz and Weber, 1976; Roberts et al., 1979).

This comment reminds me that you probably need to discuss more about the main controlling geological facts at your S Atlantic margin as well – to make sure that one understands why there is such a big difference between the two chosen case margins: e.g., thicknesses and geometries of sediments – very different, volumes and geometries of high-velocity bodies – very different, geometry of thinning in the crystalline continental crust – very different.

***Interpretation and Discussion:***

p. 17, line 10: write "field" instead of "filed"

p. 21, lines 21-23: It is difficult to make a claim that other scholars make only such simple assumptions is you read papers like Hutchinson (1085), Evans et al. (1991), Person and Garven (1992), Bertotti and ter Voorde (1994), ter Voorde and Bertotti (1994), Gvirtzman et al. (1997), Mancktelow and Grasemann (1997), Ehlers and Chapman, 1999, Lin et al. (2000), Ehlers et al. (2001, 2003), Armstrong et al. (2003), Green et al. (2004), Coolbaugh et al. (2005), to name a few, where people study temperature profiles changing in space and time (and, sometimes, not just using a heat conduction approach but also an added heat convection due to fluid flow).

The same applies to p. 21, lines 24 and 26-28. Make sure that you show the awareness of the knowledge brought by others.

***Conclusions:***

p. 22, lines 8-11: Don't you want to discuss the effect of oceanic crust transferring some heat into the adjacent distal margin? You have two case examples with dramatically different thickness of oceanic crust and one of the examples has a transform margin segment in it (that one should have done something to the thermal history of the continental margin, as we can see from publications such as Nemcok et al. (2012), Henk and Nemcok (2016)).

**Reference list:**

Not all citations are ordered alphabetically (see p. 24, Noack and Naeser).

---

## Author Comment (AC2) · 25 Dec 2017

Dear members of the Editorial Board,

First of all, we would like to show our deep appreciation towards the reviewer for reading our manuscript and providing us with the valuable suggestions and the detailed comments.

In response to the reviewer's comments, we tried to cover all mentioned points to enhance the manuscript and to clarify our work. We also have included supplementary

material to our manuscript in response to reviewers' requests.

Please find enclosed our detailed answers to the reviewer's comments and a marked-up manuscript version to track changes in the supplement to this reply. We hope the revised version of our manuscript finds your approval.

With best regards on behalf of all co-authors,

Ershad Gholamrezaie

Please also note the supplement to this comment:
https://www.solid-earth-discuss.net/se-2017-86/se-2017-86-AC2-supplement.pdf
* * *

---

## Author Response (AR1)

Dear members of the Editorial Board,

First of all, we would like to show our deep appreciation towards the reviewers for reading our manuscript and providing us with the valuable suggestions and the detailed comments.

In response to the reviewers' comments, we tried to cover all mentioned points to enhance the manuscript and to clarify our work. We also have included supplementary material to our manuscript in response to reviewers' requests.

Please find enclosed our detailed answers to the reviewers' comments and a marked-up manuscript version to track changes in the supplement to this reply. We hope the revised version of our manuscript finds your approval.

With best regards on behalf of all co-authors,
Ershad Gholamrezaie

**Authors' reply to RC #1**

**General comments**

In this paper "Variability of geothermal gradient across two differently aged continental volcanic passive margins: The Southwest African and the Norwegian margins", Gholamrezaie et al. present the results of two 3D models of the conductive thermal state in magma-rich rifted margins. They propose that the geothermal gradient is highly variable in time and from one margin to another: (1) in the amplitude of the geothermal gradients, (2) in the lateral distribution across the margin, (3) in the sedimentary basins. One of the main controlling parameter is the age of the breakup and thus the age of the oceanic lithosphere.

The authors stress that this evolution of the geothermal gradient with time, its nonlinearity with depth and its spatial variability has major implications for the calculation of paleo-temperatures and paleo-elevations (as very simple and constant paleo-gradient are used).

I think the manuscript can be improved by clarifying some points and by some additional discussions, all of which is discussed in detail below. In particular, the figure captions could be improved and more recent references could be added. In summary, this paper describes topics and an area of interest to a large audience and will be a very nice contribution to Solid Earth Discussion after minor revision.

We thank the reviewer for the constructive critics and have done our best to implement the given suggestions.

Please note that the page-line numbers in the answers refer to the marked-up version of the manuscript.

**Specific comments**

RC1-1:
I would suggest adding recent references in the geological settings of the two margins. Both area are widely covered by a substantial number of publications and only 4 references are more recent than 2010.

This is of course correct and we follow the reviewer's suggestion. The following publications have been added as recent references to the geological background of the two margins:

[revised manuscript text omitted]

For both margins, lithosphere-scale structural models and results from simulations of the steady-state conductive thermal field have been published (Scheck-Wenderoth and Maystrenko, 2008; Maystrenko et al., 2013). Though these thermal models have been produced using roughly the same workflow, there are specific differences with regard to their parametrization in response to the individual resolution and availability of data on thermal properties (Table1). With this study we concentrate on spatial variations of the present-day thermal field in response to first order differences in structural setting and related distribution of lithological units and their thermal rock properties."

**RC1-2:**
The authors state that the Norway margin is not in an equilibrated thermal state. How does this result agree with the calculation of a steady-state (i.e. equilibrated) conductive thermal model? Maybe this could be discussed.

Thank you for reminding this point and we also agree that the article misses a paragraph to discuss the controversy of steady-state thermal modelling for a margin which has not reached the thermal equilibrium. In the South Atlantic the thermal equilibration has been long completed, whereas the North Atlantic system has not yet reach this thermal equilibrium and we are aware of that. To explain why we think that a steady state calculation of the thermal field is still a good approximation we have added the following paragraphs to the discussion part 5.3 (page 22, line 31):

"One clear point stands out and that is the first order difference in the age of breakup at the two margins. Also, the timing of potential interaction in the oceanic opening with a mantle plume is different. For the South Atlantic, recent results (Scheck-Wenderoth et al., 2017) suggest that the breakup was not initiated by a mantle plume, but some interactions with the Tristan da Cunha hot spot may have influenced the evolution. Both the opening and the potential breakup-plume interactions were terminated at about 130 Million years b.p. Thus assuming steady-state thermal conditions today is a valid hypothesis as thermal equilibration has been achieved. This is also expressed in the thick oceanic mantle lithosphere derived from seismological and gravity data.

In contrast, the Iceland plume is next door to the Norwegian North Atlantic margin today (Steinberger and Torsvik, 2012) and breakup is significantly younger (55 Million years b.p.). Given the time-thermal constant of the lithosphere thermal disturbances older than 65 Ma would largely have equilibrated (Turcotte and Schubert, 2014). This implies that the thermal consequences of breakup should have declined after 55 Million years and that the system is at least close to thermal equilibrium today. Though this consideration does not account for additional dynamic thermal effects induced by the Iceland plume we use a steady-state thermal model (Scheck-Wenderoth and Maystrenko, 2008) to assess the deep thermal field for two reasons: (1) We aim to base our comparison of the two margins on the same method and (2) Both models are consistent with observed temperatures in the upper few km (e.g., Channell et al., 2006) and with surface heat flow (e.g., Ritter et al., 2004).

We would additionally like to point out that the potential error related to the steady-state assumption would, in the worst case, cause an underestimation of the mantle heat input, but not the other way round. To reach the high observed shallow temperatures with a transient simulation, while respecting at the same time the thermal lithosphere base indicated by seismology and gravity, would either require a higher temperature at the base of the lithosphere (if the process is at an early stage) or a late stage in thermal equilibration (close to steady-state).

A further discussion point relates to the nature of the lower crustal body. Depending on which lithology is assumed for the latter, higher (mafic underplate: gabbro) or lower (serpentinized mantle) thermal conductivities and different amounts of radiogenic heat production would result. Autin et al. (2016) have examined the thermal implications of these different compositions for lower crustal bodies at the Argentine magma rich margin of the South Atlantic. They found that apart from the serpentinite model being colder, the thermal effects of gabbro and eclogites would be similar."

RC1-3:
There is no reference for the statement: "One of the typical characteristics of volcanic passive margins is an extremely thinned continental crust (≈ 5 km in the distal margin)" (p. 5 line 9). Actually, I would rather state that it is a characteristic of magma-poor rifted margins. Hyper-extended continental crust in magma-rich settings is proposed, but with consequent magmatic additions that increase the thickness of the crust.

Many thanks for pointing this out! We totally agree with your comment that the text is confusing to distinguish the differences between magma-poor and -rich margins. Therefore, we have rephrased the text as follow (page 5, line 27):

"One of the typical characteristics of magma-rich passive continental margins is an only moderately thinned at the proximal margin (compared to magma-poor margins) whereas crustal thinning at the distal margin is significant. As their parts of the continental crust are replaced by lower crustal bodies, the remaining ordinary crystalline crust is thinned to a few km. These lower crustal bodies are usually characterized by high p-wave velocities of more than 7.3 km/s (White et al., 1987; Talwani and Abreu, 2000; Franke, 2013)."

**RC1-4:**

Figure 10 illustrates the evolution of the mean thermal gradient with increasing depth intervals. Maybe it could be helpful for the reader to represent the more classical "temperature vs depth" graph for each margin. It would be interesting to represent it also for each domain. While it is not well constrained, the author could also propose an evolution of this gradient through time in order to better illustrate the time variability.

We do not agree that the classical "temperature vs depth" graph would better illustrate the difference, as this graph varies for different parts of the margins. We prefer to illustrate the changes in map view as in response to the 3D nature of the margin configurations also the "temperature vs depth" varies. The temperature distribution maps for the two settings have been published (Norway: Scheck-Wenderoth and Maystrenko, 2008; SW Africa: Maystrenko et al., 2013) and validated with temperatures measured in deep wells and with surface heat-flow data (Ritter et al., 2004; Channell et al., 2006; Hartwig et al., 2010). For our analysis of these 3D variations, we have created maps that show the temperature distribution at the different depth levels for each margin to illustrate the respective "temperature vs depth variations" for all domains (complementary to the geothermal gradient maps in Figs. 8 & 9). These temperature vs depth maps will be added to the article as supplementary data.

We are thankful for raising the discussion of the geothermal gradient evolution through time. We agree that it is a very interesting topic and will be helpful for better illustration of time variability. Simulation of the thermal history however should also consider the coeval deformation and should be validated by paleo-temperature indicators. This, in turn, is a topic worth an extra study and is beyond the scope of this paper. Moreover, three-dimensional thermo-dynamic models are still a numerical challenge with respect to resolution and computation time and so far no data-based full three-dimensional lithosphere-scale cases have been presented.

Our goal here is to assess and analyze present-day thermal characteristics in three dimensions using an observation-based lithosphere-scale approach, but also considering physical principles of heat transport. The only point where we invoke time is related to the age difference of the two margins compared and to the hypothesis tested if this difference is expressed in the shallow thermal field.

**RC1-5:**

In the conclusions, you mention the influence of the Iceland plume. However, the role of plumes and their relative timing with the breakup (wide debate) are not discussed at all in the paper. A brief paragraph could be added.

The following paragraph has been added to discussion section part 5.3 (page 23, line 23):

"Proximity to Iceland mantle plume might also be effective in causing high geothermal gradients in the oceanic crustal domain of the Norwegian margin. The North Atlantic breakup was possibly initiated by the abnormally hot mantle of the Iceland plume (White, 1989; Skogseid et al., 1992; Gernigon et al., 2004, 2006; Parkin and White,

2008) activated approximately 5 million years earlier than the continental breakup (Saunders et al., 1997). While some studies have shown that the Iceland plume propagated northward (e.g., Ruedas et al., 2007; Steinberger et al., 2015), seismic tomography (Rickers et al., 2013) suggests lateral movement of plume material in addition to the parallel propagation along the mid ocean ridge. Moreover, 3D thermomechanical models (Koptev et al., 2017) suggest that plume-related thermal perturbations such as hot mantle lateral flows may result in topography at the Norwegian passive margin with long wavelength variations onshore and short wavelength variations offshore. To quantify such effects, future studies need to consider observation-based configurations, but also implement the physics of processes controlling mass and energy transport related to mantle flow dynamics."

**Technical corrections, minor suggestions**

**RC1-6:**
Title: The margin community would rather use "magma-rich rifted margins" than "volcanic passive margins" but this is not a problem for the understanding of the paper.

We have rephrased the title to the reviewer's suggestion to be more clear to our audience:

"Variability of the geothermal gradient across two differently aged magma-rich continental rifted margins of the Atlantic Ocean: The Southwest African and the Norwegian margins"

**RC1-7:**
Part 2.2: I did not understand the significance of "upper thermal boundary" at first. Maybe it should be defined.

We added a sentence regarding the thermal boundaries in the thermal models in Part 2.1 (page 3, line 22):

"The two considered 3D conductive thermal models (Scheck-Wenderoth and Maystrenko, 2008; Maystrenko et al., 2013) were created as a numerical solution to Eq. 2 in the steady-state condition ($\frac{\partial T}{\partial t} = 0$) and by considering lithology-dependent thermal properties (Table 1). The lower thermal boundary in these models has been fixed at the 1300 °C isotherm signifying the thermal LAB depth, whereas a constant temperature (Norway: 2 °C; SW Africa: 5 °C) has been set as the upper thermal boundary at the topography-bathymetry surface."

**RC1-8:**
Part 2.2: The order of citation of the figures in the text is not respected: figure 1 and then directly figure 4.

Done! We deleted the mentioned figures in this sentence:

"To calculate the geothermal gradient (Eq. 3), we considered "Tj" and "zj" respectively as the temperature and the elevation of a surface for which the upper thermal boundary was assigned to (Fig. 4a and 7a)."

**RC1-9:**
Part 2.2, line 26: Please explain why it is more relevant to use increasing thicker intervals of calculation with depth.

We have chosen this way of illustrating the depth evolution of the geothermal gradient to make our assessment of average geothermal gradient variation comparable to the observation-derived geothermal gradient variation. In practice geothermal gradients are often calculated from surface heat flow and bottom-hole temperature measurements. As therefore bottom-hole temperatures depend on the absolute depth of the drilled well, the derived average geothermal gradients vary accordingly. Our goal was to show: (1) that there is no such a thing as one average geothermal gradient and (2) that the latter is subject to variation in response to depth and structural heterogeneity. We have added this explanation in Methods (page 5, line12).

**RC1-10:**
Part 4.3, line 28: please remind the reader the ages.

Done!

"The 65 Ma year difference in the age of the oceanic crust (SW Africa: 130 Ma; Norwegian: 55 Ma; see Table 1) and the related age-controlled depth of the thermal LAB (Fig. 4a and 7a) would be a reasonable explanation for this difference within the oceanic crustal domain of the two differently aged margins."

**RC1-11:**
Part 5, line 3: add an "s" to "margin".

Done!

"According to our results, the calculated geothermal gradients reveal variations both laterally and with depth for the two different passive margins (Fig. 8 and 9)."

**RC1-12:**
Part 5.3, line 11: Please remind the reader all the mentioned hypothesis.

Done!

We added the mentioned hypothesis to the sentence as follow:

"With respect to the hypothesis formulated in the introduction that the present-day thermal field at passive continental margins is determined by lithospheric mantle

characteristics, our results suggest that considering variations in the crust alone is not sufficient to properly assess the present or past thermal configuration of passive continental margins.

**RC1-13:**
Part 6, line 12 "in" instead of "of".

Done!

"Therefore, the Norwegian model is significantly hotter than the SW African model in the oceanic crustal domain and in the distal margin."

**RC1-14:**
All maps: a short title on each map would greatly help the reader.

Done!

**RC1-15:**
Figure 2: Does the sediment thickness map includes Proterozoic sediments?

Yes! It was mentioned in the text (Part 3.2, line 25): "Onshore, the model also differentiates upper Proterozoic sediments (Owambo and Nama basins: (Miller, 1997; Clauer and Kröner, 1979))."
Additionally, we rephrased the figure caption to include these sediments.

**RC1-16:**
Figure 8 and 9: Abbreviations should be explained in the captions. Precambrian basins could be highlighted. The color scale changes for each map: would it be interesting to keep it the same for each interval?

The Precambrian basins are highlighted in the new figure 8, and within the new captions of both figures the abbreviations are explained. The color scales are set to the same color scale for all depth intervals for each figure.

"**Figure 8.** Geothermal gradient [°C/km] at SW African margin: the gradient calculated as the temperature differences between the uppermost surface (upper thermal boundary) and the corresponding temperature distribution at (a) 1, (b) 2, (c) 3, (d) 4, (e) 5, and (f) 6 km below the uppermost surface (COB: Continent-Ocean Boundary; Cretaceous-Cenozoic basins: WB: Walvis Basin, LB: Lüderitz Basin, OB: Orange Basin; Precambrian basins: OwB: Owambo Basin, NB: Nama Basin; UTM: WGS84, 33S)."

"**Figure 9.** Geothermal gradient [°C/km] at Norwegian margin: the gradient calculated as the temperature differences between the uppermost surface (upper thermal boundary) and the corresponding temperature distribution at (a) 1, (b) 2, (c) 3, (d) 4, (e) 5, and (f) 6 km below the uppermost surface (COB: Continent-Ocean Boundary; Cretaceous-Cenozoic basins: VB: Vøring Basin, MB: Møre Basin; UTM: WGS84, 33N)."

**RC1-17:**
Figure 10: It would be more logical to attribute the blue color to SW Africa and the red color to Norway as it is hotter.

Done!

**Authors' reply to RC #2**

This paper represents an interesting addition to the research focused on cooling histories of continental margins and their controlling factors. Although I have enjoyed reading it, it can be further improved to reach good quality. Below are my key points suggesting revisions of specific items. Given the extent of suggested revision, it can be classified as moderate.

We thank the reviewer for the constructive critics and have done our best to implement the given suggestions.

Please note that the page-line numbers in the answers refer to the marked-up version of the manuscript.

**General comment:**

Paper needs to demonstrate more awareness of previous work. While the previous work on specific chosen margin examples is cited a bit better, the paper seems to work in relative isolation from studies done on thermal histories and their controlling factors.

We have added more references to previous works, see also answers to RC#1, RC1-1, RC1-2, RC1-4, and RC1-5.

Concerning thermal histories, we do not go into much detail as we do not reconstruct thermal histories. The point we want to make is to raise awareness in the context of paleo-thermal conditions. Our goal is to show that even across one single margin, thermal gradients vary significantly today and margins of different age can even display different general trends in temperature variations. We have added this in a hopefully more understandable way in the "Introduction" and "Exploited models".

**Specific items:**

**Abstract:**

**RC2-1:**
Line 11 – make a full term and place LAB into parentheses. Otherwise your reader has to wait until she gets to page 7 to understand what you mean.

Done!

**Introduction:**

**RC2-2:**
p. 1, lines 16 and 21: Order your citations according to the publication year. Do this in the entire manuscript.

Done!

**RC2-3:**
p. 1, line 21: Use original references instead of relatively modern ones wherever applicable to honor the scholars who came up with certain idea originally. Do this in the entire manuscript.

Good point, done, see also answer RC1-3 to RC#1.

We have changed the paragraph as follows (page 2, line1):

"The lithospheric thermal field generally depends on the thermal thickness and the thermal properties of the lithosphere. This has been deduced from continental crustal geotherm (Pollack, 1986; McKenzie and Bickle, 1988; Rudnick and Nyblade, 1999; Kaminski and Jaupart, 2000; Artemieva and Mooney, 2001; Artemieva, 2006; Jaupart and Mareschal, 2007; Mareschal and Jaupart, 2013) and from plate cooling models explaining oceanic heat flow pattern and seafloor depth evolution (Parsons and Sclater, 1977; Johnson and Carlson, 1992; Stein and Stein, 1992; Goodwillie and Watts, 1993; DeLaughter et al., 1999; Watts and Zhong, 2000; Crosby et al., 2006; Crosby and McKenzie, 2009). There is a consensus that conduction is the main heat transfer mechanism in the lithosphere and generally controlled by (1) the heat input from larger mantle depths, (2) the heat internally produced in the lithosphere by the decay of radioactive elements, and (3) the thermal conductivity of different lithospheric layers (Summary in Allen and Allen, 2005; Turcotte and Schubert, 2014)."

**RC2-4:**
p. 2: When you are introducing the thermal history of both the oceanic crust and lithospheric mantle, you need to use the knowledge from the pioneers of this research: Parsons and Sclater (1977), Stein and Stein (1992), Goodwillie and Watts (1993), Watts and Zhong (2000).

Done, see above!

**Method:**

**RC2-5:**
p. 3: Tab. 1 needs a bit of explanation either in figure caption or, preferably, in the text. The reason for this is that you are making a claim that both margins have a very similar configuration of crust (p. 2, lines 29-30) but Tab. 1 indicates a large difference in characterizing the average thermal conductivities of oceanic crust, and a distinct

difference between the conductivities of high-velocity bodies. Such a difference should have an impact on the thermal history of these two study areas.

Many thanks for pointing out these differences. Actually, the distinct difference between the thermal conductivity of high-velocity bodies was a typo. We have corrected the mistake and changed the value from 2.3 to 2.6 [W/mK] for the Norwegian model (Scheck-Wenderoth and Maystrenko, 2008) which is similar to corresponding value of the SW African model. See also answer to RC2-11.

RC2-6:
p. 4, line 24: Make sure that your figures are cited in ascending order in your text. Here you are making a jump from Fig. 1 directly to Figs. 4 and 7.

Done! We deleted the mentioned figures in this sentence:

"To calculate the geothermal gradient (Eq. 3), we considered "Tj" and "zj" respectively as the temperature and the elevation of a surface for which the upper thermal boundary was assigned to (Fig. 4a and 7a)."

RC2-7:
p. 4, lines 25-27: It would be better to compare "apples and apples". Instead of comparing thermal gradients of 1 km-thick layers, you are comparing those of layers which are progressively 1km thicker than each previous one. I know that you can still see the downward-decreasing gradient using this approach, but aren't we supposed to compare the most directly comparable things when we do the research based on a comparison?

We understand your point regrading comparing "apples and apples", however, our objective is to compare the variation of the geothermal gradient with increasing depth interval. We have chosen this way of illustrating the depth evolution of the geothermal gradient to make our assessment of average geothermal gradient variation comparable to the observation-derived geothermal gradient variation. In practice geothermal gradients are often calculated from surface heat flow and bottom-hole temperature measurements. As therefore bottom-hole temperatures depend on the absolute depth of the drilled well, the derived average geothermal gradients vary accordingly. Our goal was to show: (1) that there is no such a thing as one average geothermal gradient and (2) that the latter is subject to variation in response to depth and structural heterogeneity. We have added this explanation in "Methods" (page 5 line12).

RC2-8:
p. 5, Fig. 1 caption: Here you wrote a caption, which could make an impression that you are calculating thermal gradients for six 1km-thick layers. Make sure that this caption is in accordance with your text on p. 4, lines 25-27.

We rephrased the caption as follow:

Also we have added to the text (page 5 line12, see also above and answer RC1-9 to RC#1):

"We have chosen this way of illustrating the depth evolution of the geothermal gradient to make our assessment of average geothermal gradient variation comparable to the observation-derived geothermal gradient variation. In practice geothermal gradients are often calculated from surface heat flow and bottom-hole temperature measurements. As therefore bottom-hole temperatures depend on the absolute depth of the drilled well, the derived average geothermal gradients vary accordingly."

**Exploited models:**

**RC2-9:**
p. 5, lines 9 and also 11: You can use older original references, rather than a random choice of younger ones. It would show your command of literature and understanding, which studies brought the original knowledge and which ones were just developing it further. Suggestions: use some of the articles by Huismans and Beaumont and some of the articles with Manatschal co-authoring, for example.

True that the work of Huismans & Beaumont, and Manatschal et al. have brought new conceptual advances in our understanding of passive margins. We added this information and the following references (part 3.1, page 5 and 6):

Huismans, R. S. and C. Beaumont (2008). Complex rifted continental margins explained by dynamical models of depth-dependent lithospheric extension. Geology 36(2): 163-166.
Lavier, L. L., & Manatschal, G. (2006). A mechanism to thin the continental lithosphere at magma-poor margins. Nature, 440(7082), 324-328.

**RC2-10:**
p. 6, lines 3 and 10-12: Use just 2-3 references for specific knowledge item. You will save some space. Make sure that you choose the original ones for the idea.

Done!

**RC2-11:**
p. 6, line 16: Here you are describing a similar character of both margins. However, this could be a good place to discuss those differences in thermal conductivities from Tab. 1 to lay down the groundwork for your later discussion about reasons for 2 different thermal histories. Here you can also touch on different thickness and distribution of sedimentary cover …etc. Honestly, when I look at your Fig. 11 a, the two margins look rather different. Thicknesses of adjacent oceanic crusts are very different. Thicknesses

and geometries of sediments – very different, volumes and geometries of high-velocity bodies – very different, geometry of thinning in the crystalline continental crust – very different. If you do not make a thorough comparison, your reader may think that you have found very similar margins where one can see what the different time for the dissipation of rifting/breakup-controlled transient does to their present-day thermal regime. However, your case requires much more thinking involved in the comparison of the two margins because the ratio of various interacting factors in control of their present-day thermal structure is different.

Thanks for making this clear. We have added the following paragraph (page 7, line29):

"There are, however also some major differences between the two margins. Major differences are related to the different times of breakup and the different post-breakup histories. The younger N-Atlantic margin is bordered by a younger and thinner oceanic lithosphere and shows a thickened oceanic crust near the continent-ocean transition compared to the S-Atlantic margin.

For both margins, lithosphere-scale structural models and results from simulations of the steady-state conductive thermal field have been published (Scheck-Wenderoth and Maystrenko, 2008; Maystrenko et al., 2013). Though these thermal models have been produced using roughly the same workflow, there are specific differences with regard to their parametrization in response to the individual resolution and availability of data on thermal properties (Table1). With this study we concentrate on spatial variations of the present-day thermal field in response to first order differences in structural setting and related distribution of lithological units and their thermal rock properties."

**RC2-12:**
p. 6, line 26: "… COB after Pawlowski (2008)…" should be rather described as "COB determined from ….this and this constraining data (Pawlowski, 2008)…"

Done! We rephrased the text as follow:
"The Continent-Ocean Boundary (COB; determined from gravity data in combination with reflection seismic and magnetic data (Pawlowski, 2008)) runs approximately along the 5 km isopach of the sedimentary fill and parallel to the coastline."

**RC2-13:**
p. 10: You also have one more problem to discuss, if you want to compare the two chosen margin examples, because they are not "apples and apples". While the S Atlantic one is a pure rifted margin, the Norwegian one has a large transform margin segment dividing the Møre and Vøring rifted margin segments. The two Norwegian margins are also characterized by being tectonically and thermally affected by multiple rifting events, instead of a single one that evolved into the breakup in S Atlantic case. How do you filter out these two effects in the case of Norwegian example to be able to compare the two case margins with respect to their controlling factors such as the LAB geometry, thermal blanketing by young sediments and thinning geometry of the continental crust?

We do only partly agree with the reviewer here. As stated before, the history itself is not the target of this study, but only a best possible description of the present-day state at, however, two differently aged margins. Therefore, we refrain from discussing the rifting and breakup history (see also answer RC1-4 to RC#1). The present-day controlling factors are derived purely based on observations of present-day structure, velocities, densities, temperatures and heat flow.

Concerning the transform character, we don't see the Norwegian Møre and Vøring margin segments as transform margins. Yes, the two segments are separated by the Jan Meyen lineament in the prolongation of an oceanic transform fault, but they look like passive margins, without any step in crustal thickness from the continent to the ocean as transform margins would display. Likewise, in the S-Atlantic a similar segmentation is present where the Walvis-, Lüderitz- and Orange basins represent different segments.

Results:

RC2-14:
p. 13, line 18: replace "theses" by "these".

Done!

RC2-15:
p. 14 and 15, Fig. 8 and 9 captions: This caption describes already a third version of your thermal gradient calculation, now letting your reader think that they are calculated at a set of six depth levels, the deepest one being 6 km deep. Make sure that your manuscript carries a unified story of your thermal gradient calculation and display.

We rephrased the caption as follow:

"**Figure 8.** Geothermal gradient [°C/km] at SW African margin: the gradient calculated as the temperature differences between the uppermost surface (upper thermal boundary) and the corresponding temperature distribution at (a) 1, (b) 2, (c) 3, (d) 4, (e) 5, and (f) 6 km below the uppermost surface (COB: Continent-Ocean Boundary; Cretaceous-Cenozoic basins: WB: Walvis Basin, LB: Lüderitz Basin, OB: Orange Basin; Precambrian basins: OwB: Owambo Basin, NB: Nama Basin; UTM: WGS84, 33S)."

"**Figure 9.** Geothermal gradient [°C/km] at Norwegian margin: the gradient calculated as the temperature differences between the uppermost surface (upper thermal boundary) and the corresponding temperature distribution at (a) 1, (b) 2, (c) 3, (d) 4, (e) 5, and (f) 6 km below the uppermost surface (COB: Continent-Ocean Boundary; Cretaceous-Cenozoic basins: VB: Vøring Basin, MB: Møre Basin; UTM: WGS84, 33N)."

RC2-16:
p. 16: The Norwegian Margin: Here you need to do more than the descriptions of geological reasons for gradient distributions that you have here. The reason is that when

you want to compare various geological reasons for such complex (and not equilibrated yet) Møre-Vøring thermal field, you need to know that:

It is the deformation history that has a controlling role on the tectonic and thermal development, as concluded from a comparison of Møre and Vøring neighbor margins in Norway (Fernandez et al., 2005). The differences of the magma-rich Vøring margin from magma-poor Møre margin are:

1) the occurrence of the extra rifting event at the beginning of the rifting history;

2) two times thicker underplated body underneath the distal margin;

3) 30 km thicker original Caledonide lithosphere;

4) a slightly smaller stretching factor;

5) larger thickness of adjacent oceanic crust; and

6) a 10 km thinner lithosphere underneath the distal margin.

These differences were attributed to different rifting histories, including the enhanced heat transfer from the oceanic crust adjacent to the Møre margin to continental crust of the Vøring margin through the contact provided by transform and occurrence of the ridge jump responsible for the separation of the Jan Mayen micro-continent initially adjacent Møre margin (Fernandez et al., 2005).

The cumulative length of rifting events at a magma-rich Vøring margin is long. The extension initiated here in late Permian and ended by Paleocene/Eocene break-up, comprising late Permian-Triassic, late Jurassic-Early Cretaceous, Late Cretaceous-Paleocene extensional events (Ziegler, 1989; Brekke, 2000; Skogseid et al., 2000; Gernigon, 2002; Van Wijk and Cloetingh, 2002).

The regional crustal stretching and subsequent crustal necking in the Vøring scenario is characterized by the last activity timing shift towards the stretching axis (Geoffroy, 1994, 2005; Schlindwein and Jokat, 1999). Unlike the S Atlantic example, in the Vøring example the crustal stretching and stretching/necking transition took about 204 Ma, although characterized by discontinuous extension. The extension initiated in late Permian in outboard locations and continued until the Paleocene/Eocene boundary in inboard locations (Ziegler, 1989; Brekke, 2000; Skogseid et al., 2000). The Paleocene was characterized by the emplacement of traps that buried the pre-existing Late Cretaceous normal fault patterns (Geoffroy, 1994; Gernigon, 2002). The transition from crustal stretching to necking then took place rather quickly, during Paleocene/Eocene transition, culminating with the break-up (Gernigon, 2002; Van Wijk and Cloetingh, 2002). While the Mesozoic stretching rate was as low as $7*10^{-16}s^{-1}$, taking place during 75 Ma (Gernigon, 2002), the Paleocene/Eocene stretching-necking transition was exceptionally fast (Hinz and Weber, 1976; Roberts et al., 1979).

This comment reminds me that you probably need to discuss more about the main controlling geological facts at your S Atlantic margin as well – to make sure that one understands why there is such a big difference between the two chosen case margins: e.g., thicknesses and geometries of sediments – very different, volumes and geometries of high-velocity bodies – very different, geometry of thinning in the crystalline continental crust – very different.

Indeed, there are variations between Vøring and Møre neighbor margins in Norway. The southern of the two (Møre) has been stretched less and has been less affected by magmatism, but has a high velocity lower crustal body. We would like to repeat that it was not the scope of this paper to unravel the rifting history, but to assess the snap shot of present-day deep temperature distribution. We agree with most of the points listed by the reviewer but a sound contribution to these points certainly would require to constitutively study the rifting history. See also detailed answer RC1-4 to RC#1.

However, one clear point stands out that there is a first order difference between the N- and S- Atlantic: in the age of breakup and with respect to the timing when a mantle plume was interacting with the system. This first order difference, results in first order different thermal fields in the South and North Atlantic. In the South Atlantic thermal equilibration has been long completed, whereas the North Atlantic system has not yet reached thermal equilibrium. Thus there, the steady-state calculation is a first approximation only and we are aware of that.

Interpretation and Discussion:

RC2-17:
p. 17, line 10: write "field" instead of "filed".

Done!

"…. a valid interpretation of shallow thermal field variations across the two differently aged passive margins."

RC2-18:
p. 21, lines 21-23: It is difficult to make a claim that other scholars make only such simple assumptions is you read papers like Hutchinson (1085), Evans et al. (1991), Person and Garven (1992), Bertotti and ter Voorde (1994), ter Voorde and Bertotti (1994), Gvirtzman et al. (1997), Mancktelow and Grasemann (1997), Ehlers and Chapman, 1999, Lin et al. (2000), Ehlers et al. (2001, 2003), Armstrong et al. (2003), Green et al. (2004), Coolbaugh et al. (2005), to name a few, where people study temperature profiles changing in space and time (and, sometimes, not just using a heat conduction approach but also an added heat convection due to fluid flow).

The same applies to p. 21, lines 24 and 26-28. Make sure that you show the awareness of the knowledge brought by others.

We have rephrased the beginning of part 5.4 and added some of the suggested references to cover this fact that not all the scholars consider the simple assumptions regarding space-time variations of temperature profiles (page 24, line 5):

"More recent studies apply complex modelling approaches to simulate thermal histories considering spatial and temporal variations in temperature profiles (e.g., Person and Garven, 1992; Bertotti and Ter Voorde, 1994; Ehlers and Farley, 2003; Ehlers et al., 2003). However, these methods of thermal history reconstruction are mostly based on paleotemperature indicators, that experience irreversible structural changes when passing through a certain temperature window (Allen and Allen, 2005; Naeser and McCulloh, 2012), but do not consider observation based on three-dimensional structural settings."

Conclusions:

RC2-19:
p. 22, lines 8-11: Don't you want to discuss the effect of oceanic crust transferring some heat into the adjacent distal margin? You have two case examples with dramatically different thickness of oceanic crust and one of the examples has a transform margin segment in it (that one should have done something to the thermal history of the continental margin, as we can see from publications such as Nemcok et al. (2012), Henk and Nemcok (2016)).

Indeed, heat is transferred from the oceanic domain to the adjacent distal margin as a natural consequence of the 3D heat transport. Thereby the local structural configuration and the related variable distribution of thermal rock properties leads to very specific thermal pattern, be it in the area of "normal" or transfer segments of the margin. Accordingly, our results are consistent with earlier studies as the one mentioned by the reviewer. We added the following sentence to the discussion (part 5.2: page 21, line 9), as we do not find it optimally placed in "Conclusions":

"Heat is transferred from the oceanic domain to the adjacent distal margin as a natural consequence of the 3D heat transport. Thereby the local structural configuration and the related variable distribution of thermal rock properties leads to very specific thermal pattern, be it in the area of "normal" or transfer segments of the margin. Accordingly, our results are consistent with earlier studies analyzing the transition from the oceanic to the continental domain (Nemcok et al., 2012; Henk and Nemcok, 2016)."

Reference list:

RC2-20:
Not all citations are ordered alphabetically (see p. 24, Noack and Naeser).

Done!
The reference list has been edited to a correct alphabetic order.

[revised manuscript text omitted]